# Natural Products as Inducers of Non-Canonical Cell Death: A Weapon against Cancer

**DOI:** 10.3390/cancers13020304

**Published:** 2021-01-15

**Authors:** Giulia Greco, Elena Catanzaro, Carmela Fimognari

**Affiliations:** Dipartimento di Scienze per la Qualità della Vita, Alma Mater Studiorum—Università di Bologna, Corso d’Augusto 237, 47921 Rimini, Italy; giulia.greco9@unibo.it (G.G.); elena.catanzaro2@unibo.it (E.C.)

**Keywords:** natural products, cancer, non-canonical cell death, ferroptosis, necroptosis, pyroptosis, in vitro studies, in vivo studies

## Abstract

**Simple Summary:**

Anticancer therapeutic approaches based solely on apoptosis induction are often unsuccessful due to the activation of resistance mechanisms. The identification and characterization of compounds capable of triggering non-apoptotic, also called non-canonical cell death pathways, could represent an important strategy that may integrate or offer alternative approaches to the current anticancer therapies. In this review, we critically discuss the promotion of ferroptosis, necroptosis, and pyroptosis by natural compounds as a new anticancer strategy.

**Abstract:**

Apoptosis has been considered the main mechanism induced by cancer chemotherapeutic drugs for a long time. This paradigm is currently evolving and changing, as increasing evidence pointed out that antitumor agents could trigger various non-canonical or non-apoptotic cell death types. A considerable number of antitumor drugs derive from natural sources, both in their naturally occurring form or as synthetic derivatives. Therefore, it is not surprising that several natural compounds have been explored for their ability to induce non-canonical cell death. The aim of this review is to highlight the potential antitumor effects of natural products as ferroptosis, necroptosis, or pyroptosis inducers. Natural products have proven to be promising non-canonical cell death inducers, capable of overcoming cancer cells resistance to apoptosis. However, as discussed in this review, they often lack a full characterization of their antitumor activity together with an in-depth investigation of their toxicological profile.

## 1. Introduction

Historically, cell death has been classified into two main categories: accidental [i.e., non-programmed cell death (PCD)] and PCD. Apoptosis and autophagy are both forms of PCD, while necrosis, instead, has been for a long time considered as a non-physiological process that occurs as a result of infection or injury [1]. However, in recent years accumulating evidence increasingly pointed out that various non-apoptotic forms of PCD, also called non-canonical, can be triggered independently of apoptosis or when the apoptotic process appears to be altered or inhibited [1,2,3]. Non-canonical cell deaths differ from the apoptotic process not only in morphological, but also in biochemical terms, and include various PCD pathways such as ferroptosis, necroptosis, and pyroptosis, which, on the contrary, can share the lytic nature with necrosis [1,4,5].

Nature is a never-ending source of preventive and curative agents, used since ancient times in traditional medicines to prevent and cure many human diseases [6]. Nature still continues to represent an inexhaustible source of pharmacologically active compounds, especially in the anticancer therapy field. Indeed, of the 185 new anticancer drugs discovered between 1981 and 2019, about 65% are natural or natural-based compounds [7]. Most of the discovered natural anticancer drugs originate from plants. There are about 250,000 plant species used for medicinal purposes, which played a crucial role for the treatment of different human diseases, according to the World Health Organization (WHO) [8]. Medicinal plants contain numerous compounds, known as primary and secondary metabolites [8]. By isolating bioactive compounds as drugs, developing bioactive compounds as semi-synthetic lead compounds, or using the whole or part of the plant, medicinal plants have been, and are still being used, as therapeutic antitumor agents [8]. The most effective drugs currently used in the oncological field are, among others, the vinca alkaloids vincristine and vinblastine, etoposide, paclitaxel, topotecan, and irinotecan, which all originate from terrestrial plants [9]. Interestingly, although until a few years ago apoptosis was the anticancer mechanism of action described for these compounds, it has been shown that some of them also induce non-canonical cell deaths [10,11]. Many other different natural compounds were thus explored and identified as promoters of non-canonical cell death.

The aim of this review is to highlight the antitumor effects of natural products as ferroptosis, necroptosis and pyroptosis inducers, and to critically analyze the limitations and challenges associated with the development of non-canonical cell death-based anticancer strategy. Although the activation of other kinds of non-apoptotic PCD, such as autophagy, anoikis, paraptosis, partanathos, netosis, or entosis could represent new promising mechanisms for the prevention or treatment of cancer, those pathways are not characterized yet. Moreover, the ability of natural compounds to trigger them is not substantial. For this reason, we focused our attention only on necroptosis, ferroptosis, and pyroptosis, for which an extensive set of information allows a comprehensive analysis. In particular, the most characterized compounds will be analyzed in detail, while the others will be included in the tables. Most of the natural products inducing non-canonical cell death have been studied in vitro. Only for some of them there are in vivo studies. Tables reporting in vitro studies are in the main text, while tables reporting in vivo studies as well as the effects of natural inducers of non-canonical cell death used in association are included in the Appendix A.

## 2. Ferroptosis

Ferroptosis, firstly discovered by Dixon et al. in 2012 [12], is a non-canonical cell death characterized by an iron-dependent accumulation of lipid reactive oxygen species (ROS), which leads to cell demise [13]. Ferroptosis differs from any other form of regulated cell death. Morphologically, it does not involve any typical apoptotic feature; it is not characterized by cytoplasmatic swelling or disruption of cell membrane, as in necrotic cell death; the formation of typical autophagic vacuoles is not observed [12]. Ferroptotic cells, instead, are morphologically characterized by a distinct shrinkage of mitochondria with enhanced membrane density and decrease/depletion of mitochondrial cristae [12].

Ferroptosis is caused by compounds able to antagonize glutathione peroxidase 4 (GPX4) in a direct way or through the inhibition of X_c_^−^ system. X_c_^−^ system is an amino acid antiporter responsible for intracellular transport of extracellular cystine by exchanging intracellular glutamate [14] (Figure 1). Once inside the cells, cystine is reduced to cysteine, an essential substrate for glutathione (GSH) synthesis [15]. Hence, the inhibition of X_c_^−^ system alters GSH biosynthesis, reducing the antioxidant activity of glutathione and selenium-dependent GPXs [16,17,18]. Among GPXs, GPX4 is the only one able to reduce hydrogen peroxides or organic hydroperoxides into water or corresponding alcohols by converting GSH into oxidized glutathione (GSSG) [19,20] (Figure 1). Then, the inhibition of GPX4, through direct or indirect mechanisms, leads to lipid ROS accumulation and activates the ferroptotic cell death cascade [12,21,22] (Figure 1).

Iron-dependent accumulation of lipid ROS can occur through non-enzymatic and/or enzymatic lipid peroxidation. Non-enzymatic lipid peroxidation, also called lipid autoxidation, consists in a free radical-driven chain reaction where ROS initiate the oxidation of polyunsaturated fatty acids (PUFAs). Within an autocatalytic process, autoxidation can be propagated leading to membrane destruction, and subsequent ferroptotic cell death [23]. Enzymatic lipid peroxidation is mostly driven by lipoxygenases (LOXs). LOXs, through their dioxygenase activity, catalyze oxygen insertion into PUFAs membrane, generating different lipid hydroperoxides (LOOH), which can start the autocatalytic process of lipid autoxidation mentioned above [22].

If the link between lipid metabolism and ferroptosis induction is well known, how lipid peroxidation leads to ferroptotic cell death is not clear yet. Two mechanisms have been hypothesized. The first hypothesis is that lipid hydroperoxides, produced by PUFAs peroxidation, generate reactive toxic products, i.e., 4-hydroxy-2-nonenal (4-HNE) or malondialdehyde (MDA), which consequently inactivate different survival proteins, leading to ferroptosis [24]. The second hypothesis is that extensive phospholipids peroxidation leads to structural and functional modifications of cellular membrane [23].

### Natural Compounds as Ferroptosis Inducers

Several natural compounds, alone or in combination, have been found to induce ferroptosis in different in vitro (Table 1 and Appendix A) and in vivo (Appendix A) cancer models.

Amentoflavone is a flavonoid mainly found in *Selaginella tamariscina* (P. Beauv.) Spring and in other species of *Selaginella*, as well as in many other plant species [55]. Amentoflavone exhibits anticancer effects in several tumor cells by inducing apoptosis, autophagy and ferroptosis, and by inhibiting cell-cycle progression [27,56,57,58,59,60,61]. In U251 and U373 glioma cell lines and in a glioma xenograft model, but not in normal human astrocytes, it triggered ferroptotic cell death by reducing GSH and ferritin heavy chain (FTH) intracellular levels, thus leading to the accumulation of lipid ROS and malondialdehyde (MDA), a PUFAs oxidation product, and subsequent cell death [27] (Table 1 and Appendix A). Hence, amentoflavone induces ferroptosis through the rupture of iron homeostasis by reducing the intracellular levels of FTH, which is involved in the intracellular iron storage [27]. Interestingly, both in vitro and in vivo, amentoflavone induced the degradation of FTH by activating autophagy via AMPK (AMP-activated protein kinase)/mTOR (mammalian target of rapamycin)/P70S6K (phosphoprotein 70 ribosomial protein S6 kinase) signaling pathway, suggesting the induction of autophagy-dependent ferroptosis [27]. Autophagy is known as a potent ferroptosis enhancer. Ferritinophagy, in particular, degrades the iron storage protein ferritin and increases the release of free iron, leading to ferroptosis induction [62,63,64].

Two other natural compounds that trigger autophagy-dependent ferroptosis are dihydroartemisinin (DHA) and typhaneoside. DHA is a semi-synthetic derivative of artemisinin, a sesquiterpene lactone derived from *Artemisia annua* L. currently used as antimalarial agent, which promotes ferroptosis in glioma cells [40] and ferroptosis together with apoptosis in acute myeloid leukemia (AML) cancer cells [39] (Table 1). Typhaneoside, a flavonoid found in the extract of *Pollen Typhae,* triggered apoptotic and ferroptotic cell death in AML cancer cells [52] (Table 1). In particular, in AML cancer cells, as for amentoflavone, both DHA and typhaneoside induced autophagy-dependent ferroptosis [39,52] by raising the degradation of ferritin through ferritinophagy; moreover, autophagy inhibition mitigated ferroptosis induction by the two natural compounds [39,52] (Table 1). In another experimental setting, DHA did not trigger ferroptosis itself, but it sensitized resistant cancer cells to ferroptosis. In particular, in vitro [mouse embryonic fibroblasts (MEFs) and human osteosarcoma HT1080 cells] and in vivo (GPX4 iKO H292-xenografted female athymic nude-Foxn 1^nu^/Foxn1^+^ mice), DHA perturbed iron homeostasis leading to an increase in intracellular iron levels, which concurred to the restoration of RSL3′s and erastin’s ability to induce ferroptosis (Table 1 and Appendix A) [65].

Artesunate is another semi-synthetic derivative of artemisinin. It induces ferroptosis in pancreatic [34,36], ovarian [35], head and neck cancer (HNC) [33], T-cell leukemia/lymphoma (ATLL) [32], and in Burkitt’s lymphoma [31] through the modulation of different molecular targets (Table 1). One of these targets is the endoplasmic reticulum (ER). ER stress is a condition of oxidative stress and perturbations in the ER folding machinery provoked by the accumulation of unfolded/misfolded proteins. ER stress activates a signaling process, called unfolded protein response (UPR), in order to lessen ER stress and to restore ER homeostasis [66,67]. In DAUDI and CA-46 lymphoma cells, artesunate triggered ferroptosis and ER stress through the activation of ATF4 (activating transcription factor-4)-CHOP (C/EBP [CCAAT-enhancer-binding protein] homologous protein)-CHAC1 (glutathione-specific γ-glutamylcyclotransferase 1) pathway and PERK [protein kinase RNA (PKR)-like ER kinase] branch of UPR [31] (Table 1). As proposed by the authors [31], the upregulation of CHAC1 possessing a GSH degradation activity [68,69] probably contributes to artesunate-induced ferroptosis [31]. Besides, it is well known that ATF4 could be upregulated by the depletion of amino acids [70], such as that of intracellular cysteine caused by ferroptosis inducers through the system X_c_^−^ inhibition. Hence, artesunate might induce ER stress in Burkitt’s lymphoma cells by altering the system X_c_^−^, even if it has to be confirmed. ER stress is also involved in artesunate-induced ferroptosis in *KRas* mutant pancreatic cancer cells (PaTU8988 and AsPC-1) and in AsPC-1 xenografted BALB/c nude mice [34] (Table 1 and Appendix A). Indeed, knockdown of glucose-regulated protein 78 (GRP78), which is considered the master regulator of the UPR signaling process [71], and the inhibition of the three UPR transducers [PERK, IRE1 (inositol requiring protein-1) and ATF6 (activating transcription factor-6] [72], enhanced artesunate-induced ferroptosis in vitro and in vivo [34] (Table 1 and Appendix A). Of note, artesunate triggered ferroptosis in a most efficient way in pancreatic cancer cells carrying mutationally-active *KRas* mutations (i.e., AsPC-1) rather than in pancreatic cancer cells expressing wild type *KRas* (i.e., COLO-357 and BxPC-3) [36]. This outcome is not odd since *KRas* mutation often leads to low antioxidant ferritin and transferrin levels and increased number of transferrin receptors and may sensitize pancreatic adenocarcinomas to ferroptosis [33,73]. Still, given that *KRas* mutant tumors are hardly druggable, these results are quite auspicious [33,74,75].

The role of nuclear factor (erythroid-derived 2)-like 2 (Nrf-2) in ferroptosis is still a matter of debate. Normally, Nrf2 is kept inactivated by Kelch-like ECH-associated protein 1 (Keap-1). Under increased oxidative stress conditions, Nrf2 dissociates from Keap1, translocates into the nucleus, and starts the transcription of the so-called antioxidant responsive element (ARE)-dependent genes [76,77,78]. Most of the Nrf2 target genes are involved in the maintenance of redox homeostasis [79,80], including the regulation of system X_c_^−^ [81,82,83], and also in iron and heme homeostasis. They regulate heme-oxygenase 1 (HO-1), ferroportin, and light chain and heavy chain of ferritin (FTL/FTH1) [76,84,85]. In other words, Nrf2 activation could be considered a negative ferroptosis regulator since it endorses antioxidant elements and iron storage, and limits cellular ROS production [86]. Furthermore, since it has been shown that ferroptosis inducers capable of activating Nrf2 pathway promote cellular adaptation and survival and render cancer cells less sensitive to ferroptosis induction themselves [87,88,89], it could be thought that they cannot be considered good candidates for anticancer therapy. However, the activation of Nrf2 pathway could promote ferroptotic cell death. Shifting the focus from the antioxidant properties of Nrf2 effectors to their ability in increasing intracellular iron content, that evidence is not surprising. For instance, HO-1 is responsible for heme catabolism, which produces iron, monoxide, and biliverdin. Thus, it is plausible assuming that the Nrf2 antioxidant response cannot balance the strong iron production, which leads cells to ferroptosis [90]. Accordingly, Kwon et al. [90] demonstrated that hemin, the most prevalent heme metabolite originated by HO-1 catabolism, induced lipid peroxidation as a consequence of iron increase [90]. The opposite role of Nrf2 in ferroptosis seems to be cell-type specific [91], since the activation of Nrf2 pathway protected hepatocellular carcinoma cells against ferroptosis [87], while it promoted ferroptosis in neuroblastoma [54]. Taken together, those results support the hypothesis that Nrf2 could act as a double-edge sword. Even if further studies are needed to disentangle this knot, artesunate supports this hypothesis inducing different effects in different cell lines.

In HNC cells, but not in human oral keratinocytes and fibroblasts, artesunate decreased GSH intracellular levels and increased lipid ROS production and led to ferroptosis [33] (Table 1). However, in HNC cells and cisplatin-resistant HNC cells, artesunate activated the Nrf2 pathway [33] (Table 1), favoring the onset of ferroptosis resistance. As a matter of fact, Keap1 silencing decreased cancer cells’ sensitivity towards artesunate-mediated ferroptosis in both resistant and non-resistant cells, while Nrf2 silencing restored the ability of inducing ferroptosis [33]. In Panc-1 pancreatic cancer cells, induction of ferroptosis by artesunate was accompanied by an increase of HO-1 protein expression [36] (Table 1), which authors associated with the ability of artesunate to increase ROS levels, that in turn activates Nrf2-mediated antioxidant response. Hence, it could be postulated that artesunate induces ferroptosis in pancreatic cancer cells through the HO-1-mediated enhancement of intracellular labile iron (LIP) (i.e., ionic Fe complexes that are redox active). Promotion of ferroptosis by artesunate has been reported also in vivo (Appendix A). In Burkitt’s lymphoma xenograft model, it suppressed tumor growth by inducing lipid peroxidation [31] (Appendix A).

Withaferin A (WA) is a naturally occurring steroidal lactone derived from *Withania somnifera*, a medicinal plant used in Ayurvedic medicine [92]. In a variety of cancer cells, WA showed to exhibit anticancer activity through a plethora of mechanisms, including proteasome and cell-cycle inhibition, modulation of oxidative stress, and induction of apoptosis [92]. In neuroblastoma cells, WA promoted ferroptosis through a dual mechanism: at high dose (10 μM), WA directly binds and inactivates GPX4, thus inducing canonical ferroptosis; at lower dose (1 μM), WA targets Keap1 and activates the Nrf2 pathway, leading to an excessive upregulation of HO-1 and a subsequent LIP increase [54] (Table 1). Through these two mechanisms, WA also promoted ferroptosis and eradicated neuroblastoma xenografts in BALB/c mice [54] (Appendix A). Of note, WA outperformed the full-blown chemotherapeutic agent etoposide both in vitro and in vivo. In vitro, WA efficiently killed a panel of high-risk and etoposide-resistant neuroblastoma cells by inducing ferroptosis [54] (Table 1). In vivo, WA intratumoral administration showed the same efficacy of etoposide in suppressing tumor growth [54] (Appendix A). Most importantly, in contrast to etoposide, WA treatment also repressed neuroblastoma relapse rates in four out of five mice [54] (Appendix A). Hence, taking into account that WA-induced cell death was associated with CD45-positive immune cells infiltration in tumor tissue [54], we could speculate that WA could have activated the immune system, thus inducing an anticancer vaccine effect (Appendix A). This result may itself be a further step in demonstrating that ferroptosis possesses an immunogenic nature, especially in light of the recent confirmation that ferroptosis could promote antitumor immunity [93]. The only sore point of the study, together with the small number of animals used in the experimentation, was the toxic effect of WA observed in vivo. Since upon WA-systemic injection severe weight loss–related adverse effects were detected and given the scarce water solubility of WA, authors formulated WA-encapsulated nanoparticles (WA-NPs) [54]. WA-NPs showed the same efficacy of non-encapsulated WA in vitro (Table 1) and in vivo (Appendix A), constraining systemic side-effects induced by WA, thus allowing systemic application and an effective tumor targeting of WA [54].

## 3. Necroptosis

The term necroptosis was coined in 2005 when Degterev et al. discovered that necrostatin-1 was able to inhibit tumor necrosis factor (TNF)-induced necrosis by blocking receptor-interacting serine/threonine-protein kinase 1 (RIP1) activity [94]. Even if necroptosis is a finely cellular death mechanism, it shares the morphological features of necrosis, such as cellular rounding and swelling, cytoplasmic granulation, and plasma membrane rupture [95]. Moreover, although necroptosis is a caspase-independent cell death mechanism, it shares some initiating factors with the extrinsic apoptotic pathway [96].

The best characterized type of necroptosis is the TNFα/TNF receptor (TNFR) signaling pathway, considered as a prototype mechanism of necroptosis induction [97]. TNFα binds and activates TNFR1, which recruits TNF receptor-associated death domain (TRADD), cellular inhibitor of apoptosis 1 and 2 (cIAP1 and cIAP2), TNFR-associated factor 1 and 2 (TRAF1 and TRAF2) and RIP1 to create a membrane-signaling complex, called complex I [98,99] (Figure 2). In this complex, cIAP1/2 induces Lys63-linked polyubiquitination of RIP1, which consequently leads to the activation of canonical nuclear factor kappa-light-chain-enhancer of activated B cells (NF-kB) pathway and, eventually, cell survival [100]. Conversely, inhibition of cIAPs activity by the second mitochondrial activator of caspase (Smac)/Diablo proteins or the Smac mimetic compounds promotes the deubiquitination of RIP1 by the deubiquitinating enzymes cylindromatosis (CYLD) and A20, which both hydrolyze Lys63-linked ubiquitin chains [100]. Subsequently, RIP1 dissociates from complex I to form either cytosolic complex IIa or complex IIb, depending on the proteins content: complex IIa is formed by TRADD, RIP1, Fas-associated death domain (FADD), and caspase-8; complex IIb is formed by RIP1, FADD, and caspase-8, but it does not contain TRADD. While in the complex IIa activation of caspase-8 is independent from RIP1 kinase activity, in complex IIb, where TRADD is not present, RIP1 kinase activity is required for caspase-8 activation and induction of RIP1-dependent apoptosis [100]. However, complex IIa and IIb are both capable of inducing apoptosis or necrosis depending on cell status. Indeed, when caspase-8 is inhibited, RIP1 interacts with and activates by autophosphorylation [101] RIP3, leading to the formation of a protein complex called necrosome [102] (Figure 2). RIP3, beside its activation through RIP1, could be directly activated also by other stimuli, as lipopolysaccharides (LPS), double-stranded (dsRNA), and DNA-dependent activator of interferon-regulatory factor [100]. The formation of necrosome induces the activation and phosphorylation of both RIP1 and RIP3, which subsequently phosphorylate mixed lineage kinase domain-like (MLKL) [103,104,105] (Figure 2). Then, phosphorylation of MLKL induces its oligomerization and translocation to plasma membrane, which is crucial for necroptosis execution [105,106,107] (Figure 2). To date, the effector mechanism by which necroptosis is executed is still controversial. Some studies report that oligomerized MLKL could interact with negatively charged phospholipids and create pore structures into the plasma membrane [108,109]. In contrast, others report that MLKL could induce a dysregulation of ionic fluxes in the plasma membrane [104,107].

### Natural Compounds as Necroptosis Inducers

Several natural compounds promote necroptosis in cancer cells both in vitro (Table 2 and Appendix A) and in vivo (Appendix A). 

Among all, shikonin is definitely the most characterized necroptosis inducer of natural origin. Shikonin is a naphtoquinone isolated from the root of *Lithospermum erythrorhizon* Sieb. et Zucc, *Arnebia euchroma* (Royle) Johnst, or *Arnebia guttata* Bunge [152]. It promotes necroptosis in a wide range of cancer cells, including pancreatic [134], nasopharyngeal [135,144], gastric [145], lung [136], breast [133,147,148], osteosarcoma [138], lymphoma [139], multiple myeloma [137], and glioma [140,141,142,143] (Table 2). In AGS gastric cancer cells, shikonin induced necroptosis or apoptosis in a time-dependent manner: with equal concentrations (1, 2, and 4 μM), the short-time treatment (6 h) led to necroptosis induction, while longer time treatment (24 h) led to apoptotic cell death [145]. In MCF-7 breast cancer cells, shikonin promoted necroptosis when the apoptotic machinery was inhibited [147] (Table 2). Interestingly, most of natural compounds illustrated in Table 2 induce both necroptosis and apoptosis, confirming the existing interrelation between the two cell death mechanisms. Indeed, cell fate (apoptosis *versus* necroptosis) is primarily affected by available caspase-8 and cIAP1, cIAP2, XIAP (X-linked inhibitor of apoptosis protein). Their deficiency favors necroptosis induction by suppressing RIP/RIP3 proteolytic cleavage or ubiquitination of RIP1 [153,154,155].

Besides the activation of RIP1 and RIP3 and the promotion of necrosome complex formation, the crucial event involved in shikonin-induced necroptosis is the production of ROS. Shikonin induced oxidative stress in nasopharyngeal [135], glioma [141,142,143], gastric [145], and breast cancer cells [147] (Table 2) and the increase in ROS levels was linked to necroptosis induction in some of those models. In glioma cancer cells, shikonin boosted ROS and mitochondrial superoxide generation [141,142,143] (Table 2). Inhibition of RIP1 and RIP3 reduced ROS, mitochondrial superoxide production, and cell death. Alongside, ROS increased RIP1 and RIP3 levels, showing that oxidative stress is a regulative factor in shikonin-mediated necroptosis [141] (Table 2). Moreover, in glioma cancer cells, oxidative stress triggered by shikonin led to the collapse of mitochondrial membrane potential, promoting the cytoplasmic release and the nuclear translocation of AIF (apoptosis-inducing factor) [142] (Table 2). Activated MLKL seems to be responsible for shikonin-induced mitochondria collapse, since its inhibition reduced ROS and superoxide production and AIF mitochondrial release [143] (Table 2). This hypothesis is further supported by the observed accumulation of MLKL in mitochondria and the enhanced expression of both mitochondrial and activated MLKL [143] (Table 2). Indeed, MLKL could boost the catalytic activity of PGAM5 (mitochondrial serine/threonine protein phosphatase family member 5) and bind mitochondrial-specific lipid cardiolipin [109], leading to mitochondrial fragmentation [105,156]. However, whether mitochondria are essential in necroptotic cell death is not clear yet. In mitochondria-deficient cells, as well as in cells from PGAM5^−/−^ mice, necroptosis still occurred [157,158]. Interestingly, the overproduction of ROS and/or the loss of mitochondrial potential are strictly involved not only in shikonin-induced necroptosis but also in many other natural-derived necroptosis inducers, including 2-methoxy-6-acetyl-7-methyljuglone [110,111], arctigenin [116], columbianadin [121], deoxypodophyllotoxin [122], matrine [126], pristimerin [129], resibufogenin [131], and tanshinol A [149] (Table 2), thus further confirming their pivotal role in necroptosis induction.

Shikonin confirmed its ability to promote necroptosis also in many different in vivo experimental models [135,136,138,142,143,144]. In female nude mice (authors did not state the species) [135], and BALB/c nude mouse xenograft models of human nasopharyngeal [135,144], or lung cancer [136], shikonin reduced tumor growth and increased tumor cell necrosis [135,136,144], which, in the latter model, have been associated with an increase in RIP1 expression in the tumor tissue [136] (Appendix A). In BALB/c nude mouse xenograft model of human glioma, shikonin induced the binding of MLKL with mitochondria and the subsequent release of AIF and promoted necroptosis [143] (Appendix A). In the same model, shikonin caused DNA damage [142] (Appendix A), as observed in several cancer cells in vitro [142,159,160] (Table 2), thus configuring itself as a possible mutagen compound. This aspect is certainly to be taken into account in the evaluation of the toxicological profile of shikonin, even if it is worth noting that the antitumor activity of several anticancer drugs is based on DNA-damage induction [161]. In BALB/c nude mouse xenograft model of human osteosarcoma, shikonin reduced tumor growth, increased RIP1/3 expression, and reduced lung metastasis thus suggesting an antimetastatic activity for shikonin [138] (Appendix A). However, attention must be paid since the role of necroptosis in cancer metastatization is controversial. Strilic et al. reported that tumor cells-induced necroptosis of endothelial cells promotes cancer cells extravasation and metastatization through interaction with DR6 (death receptor 6) [162], hence showing that necroptosis could promote cancer cell metastatization [161]

Berberine is the major component of different plants belonging to *Berberis* species, and many other plants including, among all, *Coptis chinensis* Franch., and *Hydrastis canadensis* L. [163]. Besides its widely documented apoptotic anticancer activity [163], berberine promoted necroptosis in ovarian cancer cells and in three patient-derived primary ovarian cancer cell lines (POCCLs) by activating RIP3 and MLKL [118] (Table 2). Berberine triggered necroptosis also in diffuse large B-cell lymphoma (DLBCL) cancer cells, where the necroptotic mechanism has been deeply investigated [119] (Table 2). In DLBCL cells, berberine promoted mitophagy-dependent necroptosis by inducing the formation of the RIP1/RIP3/MLKL necrosome complex and mRNA degradation of PCYT1A (phosphate cytidylyltransferase 1 alpha), thus reducing its expression in cancer cells [119] (Table 2). PCYT1A is an isoform of the CTP (choline phosphate cytidylyltransferase) enzyme, which is crucial for PC (phosphatidylcoline) synthesis [164]. The authors of the study showed that PCYT1A was overexpressed in 44% of the analyzed DLBCL patients and that PCYT1A overexpression occurred in parallel with the enhanced gene and protein expression of MYC [119], an oncogene mostly involved in lymphoma cell chemoresistance [165]. Moreover, MYC-induced overexpression of PCYT1A led to inhibition of necroptotic cell death in DLBCL cells [119] (Table 2). In this context, berberine effectively suppressed DLBCL cancer cells growth by inhibiting the MYC-driven downstream effector PCYT1A, and inducing mitophagy-dependent necroptosis [119], thus being eventually considered as a promising anticancer agent to treat MYC-overexpressing lymphomas.

## 4. Pyroptosis

The term pyroptosis was coined by Cookson and Brennan to describe a peculiar caspase-1-dependent, pro-inflammatory regulated form of cell death involved in *Salmonella*-infected macrophages [166]. The term pyroptosis has been drawn from the two ancient Greek words *pyro,* and *ptosis,* which respectively mean fire or fever, and collapse or demise [166]. Pyroptosis is involved in innate immune defense against pathogenic infections or endogenous risk signals through the recruitment of immune cells by pro-inflammatory cytokines [167]. Its overactivation or dysregulation can lead to autoimmune and autoinflammatory diseases [168]. Pyroptosis is closely linked to cancer, where it acts as a double-edged sword. Indeed, as an inflammatory cell death process, pyroptosis could promote tumor cell growth by different pro-tumorigenic mechanisms [168,169]; conversely, it could suppress tumors development [168] also by enhancing anti-tumor immunity [170,171,172].

Pyroptosis shares with apoptosis some morphological and mechanistic features, including DNA damage and caspase activation. For instance, caspase-1/4/5 but also caspase-3 are involved in pyroptotic cell death [173,174,175]. Morphologically, pyroptotic cells display DNA fragmentation and chromatin condensation, but in contrast to apoptosis their nucleus remains intact; moreover, pyroptotic cells are characterized by the formation of large bubbles at the plasma membrane resulting in cell swelling, and consequent plasma membrane permeabilization together with cellular osmotic lysis [176].

Depending on the different *stimuli* and inflammatory mediators, pyroptosis falls into a canonical or non-canonical cell death mechanism, which converge into the same effector system, i.e., the activation of one member of the gasdermin protein (GSDM) family [177]. In canonical pyroptosis, specifics pathogen-associated molecular patterns (PAMPs), damage-associated molecular patterns (DAMPs) or homeostasis-altering molecular processes (HAMPs) are recognized by inflammasome sensors [178] (Figure 3). An inflammasome is a multiprotein complex formed by (1) a sensor named PRR (pattern recognition receptor), (2) an adaptor protein apoptosis-associated speck-like protein (ASC), which contains a caspase-recruitment domain, and (3) caspase-1 [179]. Different types of PRRs are involved in pyroptosis including the nucleotide-binding oligomerization domain (NOD)-like receptors (NLRs), the absent in melanoma 2 (AIM2)-like receptors (ALRs), and pyrin proteins [180] (Figure 3). The most characterized NLRs in canonical pyroptosis is NLRP3 (NLR family pyrin domain-containing 3). A wide range of *stimuli*, such as pore-forming toxins, extracellular RNA, ROS, and mitochondrial DAMPs can trigger the NLRP3 cascade [181,182,183,184,185,186,187]. In turn, the activated-inflammasome sensors lead to the recruitment, directly or via ASC, of caspase-1 to form the full-blown inflammasome and drive caspase-1 activation [188,189]. Then, activated caspase-1 fosters the proteolytic maturation of pro-inflammatory precursors like pro-interleukin-1 beta (IL-1β) and pro-interleukin-18 (IL-18) and activation of gasdermin (GSDM) D (GSDMD) [190] (Figure 3). GSDMD could also be activated by bacterial intracellular lypopolisaccaride (LPS), without inflammasome involvement [169,191]. In the latter case, GSDMD is cleaved by caspases 4/5, which are the human caspase-11 murine orthologue. GSDMD cleavage leads to the release of the N-terminal fragment (GSDMD-NT) [186] (Figure 3). After its release, GSDMD-NT oligomerizes to form pores on the inner leaflet of the plasma membrane [192] causing osmotic cell swelling and the rupture of the plasma membrane with the spillage of the cellular content into the extracellular space, including the inflammatory cytokines IL-1β and IL-18 [193] (Figure 3). Besides, some pro-apoptotic chemotherapy drugs and molecular-targeted therapies promote pyroptosis through the caspase-3-dependent cleavage of GSDM E (GSDME) [175,194,195] (Figure 3). Cleavage of GSDME leads to the release of GSDME-NT (Figure 3), which possesses pore-forming activity as GSDMD-NT [175,194,195].

As mentioned above, pyroptosis and apoptosis are closely intertwined. For instance, NF-kB pathway is commonly referred to as apoptosis regulator [196,197], and has been found to trigger pyroptosis as well [198]. Indeed, as in the case of NF-kB, the same pro-apoptotic *stimulus* could, in some circumstances, provoke different cell death pathways [199]. The discriminating factor in triggering apoptosis, pyroptosis or both PCDs is the expression of GSDM in tumor cells. In tumors with low levels of GSDME, activated caspase-3 elicits apoptosis, while if the tumor expresses high levels of GSDME, caspase-3 switches its downstream pathway from apoptosis to pyroptosis or apoptosis and pyroptosis [175,200]. Of note, GSDME levels differ depending on the tumor type: low levels are detected in gastric and skin cancer, high levels in lung cancer, colorectal cancer, neuroblastoma, and melanoma. Thus, pyroptosis could be considered a tumor-type specific cell death [168,175].

### Natural Compounds as Inducers of Pyroptosis

Several natural compounds and their derivatives or analogues were found to induce pyroptosis in different cancer models, both in vitro (Table 3) and in vivo (Appendix A). 

Galangin is a natural flavonoid found in different plants including *Alpinia officinarum* Hance [214]. In glioblastoma multiforme cell lines (U251 and U87MG), galangin induced apoptosis, autophagy, and GSDME-mediated pyroptosis [207] (Table 3). Interestingly, it has been found that inhibition of autophagy enhances pyroptosis and apoptosis induction. Autophagy, promoting cell survival and blocking inflammation, could suppress inflammasome activation [215], thus limiting pyroptotic cell death. For this reason, the inhibition of autophagy can represent a strategy to favor pyroptosis, as observed in cells treated with galangin, the *Citrus* flavanoid nobiletin or alpinumisoflavone [202,210] (Table 3).

Polyphyllin VI (PPVI) is a steroidal saponin isolated from the ethyl acetate fraction of *Trillium tschonoskii* Maxim [199]. PPVI displayed anticancer effects against lung cancer cells and in an athymic nude mouse xenograft model of lung cancer through the induction of apoptosis, autophagy [198,199], and pyroptosis [198] (Table 3 and Appendix A). PPVI provoked pyroptosis by activating the NLRP3 inflammasome, responsible for GSDMD cleavage and caspase-1-dependent maturation and secretion of IL-1β and IL-18 [198] (Table 3). The PPVI-induced pyroptosis was associated with ROS generation and activation of the NF-kB pathway [198] (Table 3). Indeed, the increased expression of NLRP3 facilitates the NF-kB-mediated effective assembly of the inflammasome [216]. Hence, it could be supposed that the activation of NF-kB pathway by PPVI participates into the assembly of NLRP3 inflammasome and that PPVI-mediated ROS generation in turn activates NLRP3. Interestingly, ROS generation promoted PPVI-induced apoptosis [199], thus suggesting that the same cell death *stimulus* could activate different PCD pathways.

Notably, all the natural compounds illustrated in the Table 3 induced both pyroptosis and apoptosis, hence endorsing that the crosstalk between these two cell death pathways is really tight, as mentioned above. In PPVI-induced pyroptosis, the relationship between pyroptosis and apoptosis has been found to be the ROS-mediated activation of the NF-kB signaling pathway [198]. Another important observation is that the natural saponin dioscin induced apoptosis by activating the c-Jun N-terminal kinase (JNK)/p38 signaling pathway [206]. This means that dioscin-induced pyroptosis could be activated through the same pro-apoptotic upstream pathway triggering caspase-3 activation Thus, certain compounds’ ability to elicit pyroptosis in addition to apoptosis could be considered a potentially effective strategy to synergize their anticancer efficacy.

Although pyroptosis inducers can have an interesting role in the oncological field, pyroptosis induction should be carefully sought since it could have also a cancer promotion effect. Indeed, to treat certain tumors, such as skin cancer, inhibition of pyroptosis could be pursued. A persistent inflammatory status or alterations in inflammatory activity are implicated in skin tumorigenesis, together with the modulation of cancer progression and invasiveness by cytokines [217]. For instance, two natural compounds such as epigallocatechin-3-gallate and thymoquinone suppressed growth and migration of melanoma cells by inhibiting NLRP1 inflammasome, IL-1β-mediated secretion, and NLRP3 inflammasome, respectively [218,219]. Hence, inhibition of pyroptosis, instead of its induction, could be a potential antitumor strategy in skin cancer treatment, as in other tumor diseases where inflammation plays a key role in tumor progression.

## 5. Selective Activity of Natural Inducers of Non-Canonical Cell Death towards Cancer Cells

One of the main drawbacks of current anticancer chemotherapy is the non-selective cytotoxicity towards cancer cells, which is associated with the appearance of systemic toxicity and significant side effects [220]. However, only a few studies explored the impact of the previously described natural compounds on non-transformed cells, and often the results obtained in different studies are conflicting.

Regarding all the natural inducers of ferroptosis described in Table 1, controversial data arose about artesunate and WA.

Although several studies indicate that artesunate selectively kills cancer cells, many others showed cytotoxicity *versus* normal cells. The IC_50_ on human bronchial epithelial HBE cells after 24 h treatment was 1.38 times higher (212.48 μM) than that observed in A549 lung adenocarcinoma cells (153.54 μM) [221]. The IC_50_ on human osteosarcoma cells treated with artesunate for 48 h was about four times higher (206.3 μM) than that observed on the non-transformed counterpart hFOB1.19 human osteoblast (52.8 μM) [222]. On normal human urothelial SV-HUC-1 cells, the IC_50_ after 48 h artesunate treatment (1149.6 μM) was about one order of magnitude higher compared to those obtained in T24 and RT4 bladder cancer cells (129.7 μM and 103.2 μM, respectively), showing a remarkable selectivity of action [223]. Rho et al. compared artesunate cytotoxicity on both HNC cancer cells and normal oral keratinocytes (HOK) and fibroblasts (HOF), founding that all HNC cells succumbed to artesunate 100 μM, while almost all HOK and HOF cells survived to artesunate 50 μM. However, no data is available for 100 µM treatment [33]. After 24 and 48 h, artesunate at 33–521 μM exhibited a slight citoxicity on normal retina hTERT-RPE1 cells compared to retinoblastoma RB-Y79 cells (at least eight or nine times lower in hTERT-RPE1 versus RB-Y79 cells) [224]. Furthermore, artesunate showed cytotoxic effects on normal human and mouse/rat liver cells [225,226]. Indeed, exposure to artesunate 100 μM for 24, 48, and 60 h induced a significant cytotoxic effect on both human hepatocellular carcinoma cells (HepG2, Huh-7, and Hep3B) and normal hepatocytes (L02) [225]. Its cytotoxic effect was kept even at lower concentrations (0.5–10 μM), as shown on normal rat liver BRL-3A cells and mouse liver AML12 cells (24 and 48 h treatments) [226]. However, the authors of both studies [225,226] did not explicitly quantify the entity of these cytotoxic effects. Ishikava and colleagues reported that cell viability of HTLV-1 (human T lymphotropic virus type 1)-infected T-cell lines (MT-2, MT-4, and HUT-102) decreased time- and dose-dependently after artesunate exposure (20–60 μM for 24 and 48 h), but PBMCs (together with Jurkat and CEM leukemia cells) treated for 24 h with artesunate 5 and 10 μM were relatively resistant [32]. Once more, the authors of the study did not quantify this effect. Taken these studies together, we could conclude that artesunate is tumour-selective in an organ or cell-type way. However, the lack of objective and quantitative data of many studies makes it difficult to draw reliable conclusions.

WA revealed a similar tumour-type-dependent selectivity. It was cytotoxic at 2–10 μM on different human colon cancer cell lines: HCT-116 (IC_50_: 5.33 μM), SW-480 (IC_50_: 3.56 μM), and SW-620 (IC_50_: 5.0 μM) at 24 h treatment; however, no significant cytotoxic effect was found on normal colon epithelial FHC cells, even if the highest tested concentration was 6 μM [227]. Cell viability of Ca9–22 and HSC-3 human oral cancer cells treated for 24 h with WA 1 μM was 83.4% and 79.4%, respectively, while no cytotoxicity was recorded in HGF-1 normal oral cells [228]. Moreover, most of human fibroblasts (TIG-1 and KD) treated with WA 2 μM remained viable up to 96 h treatment, while DU-145 and LNCaP prostate cancer cells almost completely succumbed after 24 and 72 h, respectively [229]. Lastly, the IC_50_ on WI-38 normal lung cells after 24 h WA treatment was > 50 μM, versus ~10 μM recorded for A549 cells [230]. Furthermore, WA showed a considerable safe profile on PBMCs. The IC_50_ after 24 h was > 50 μM [230], and no major cytotoxic effect was recorded on both PBMCs and hematopoietic progenitor cells up to 48 h exposure at 30 μM [231]. At the same treatment time, instead, the IC_50_ on MOLT-4, Jurkat, REH, and K-562 leukemia cells was 1.52, 1.62, 3.09, and 0.58 μM, respectively [231]. In contrast, both U2OS osteosarcoma cells and TIG-3 normal fibroblast were killed by WA at doses equal to or less than 1.1 μM (treatment time was not specified) [232].

Dihydroisotanshinone I 10 and 20 μM after 48 h treatment significantly inhibited the proliferation of H460 (IC_50:_ 19.4 μM) and A549 (IC_50:_ 15.5 μM) small lung cancer cells [233]. However, cell proliferation of normal human lung fibroblasts (IMR-90) was only slightly inhibited after 24 h treatment with dihydroisotanshinone I 5 and 10 μM (no indication of IC_50_) [233].

The natural necroptosis inducer pristimerin showed a dubious selectivity of action. The IC_50_ on MCF-7 (breast carcinoma), HCT116, HepG2 (human hepatocellular carcinoma), SCC-4 and HSC-3 (human oral squamous cell carcinoma), and B16-F10 (mouse melanoma) cells after 72 h treatment with pristimerin was 7.9 μM, 9.4 μM, 7.8 μM, 12.7 μM, 2.9 μM, and 6.3 μM, respectively [234]. Instead, at the same conditions, the IC_50_ on normal lung MRC-5 fibroblast was 3.5 μM, showing that normal cells are even more sensitive to pristimerin activity than most tumour types [234]. Very similar results were obtained comparing the cytotoxic effect of pristimerin (72 h) on HL-60 cells (IC_50_: 1.31 μM) and K-562 leukemic cells (IC_50_: 3.2 μM) with that on PBMCs (IC_50_: 0.88 μM) [235]. Rodrigues et al. confirmed this trend showing an even more pronounced sensitivity of normal cells: pristimerin was cytotoxic on both HL-60 and K-562 leukemic cells (IC_50_ at 72 h: 8.8 μM and 13.6 μM, respectively) and markedly cytotoxic on PBMCs (IC_50_ at 72 h: 0.6 μM) [234]. In contrast, human breast epithelial MCF-10A cells were 2 to 3 times higher resistant to pristimerin than MDA-MB-231 breast cancer cells, in particular at 24 h [236]. To note, 1.5 to 12 h pristimerin exposure triggered necroptosis in glioma C6 cells at 2.5 μM and in U251 cells at 4.5 μM [129]. Even if Zhao and colleagues [129] did not test pristimerin cytotoxicity on the non-transformed counterpart, the cytotoxic concentrations of pristimerin for glioma C6 cells are higher and the treatment times shorter than those responsible for the cytotoxic effect on PMBCs and MRC-5 normal cells [234].

Among all the natural inducers of pyroptosis described in Table 3, only for some of them their selectivity of action towards cancer cells has been well established. Among them, the selectivity of dioscin towards tumour cells is still controversial. Treatment with dioscin 5.8 μM for 24 h reduced cell viability to about 70% in normal human pancreatic ductal epithelial cells (HPDE6-C7) compared to the 40% observed in ASPC-1 and PANC-1 pancreatic cancer cells [237]. At the same dose and treatment time (5.8 μM for 24 h), dioscin reduced normal nasopharyngeal NP69 cells’ viability to 73% compared to 40% and 33% of Panc-1- and ASPC-1-treated cells [238]. On normal cervical epithelial H8 cells treated with dioscin 5.8 μM for 24 h, the cell viability inhibition rate compared to untreated cells was 25% *versus* 80% and 54% observed on HeLa and SiHa cervical cancer cells, at the same experimental conditions [239]. Moreover, the IC_50_ on L02 hepatocytes after 48 h treatment with dioscin (13.23 μM) was more than six times higher than that observed in HepG2 cancer cells (2.38 μM) [240]. Lastly, the IC_50_ (the authors did not specify the treatment time) on NOZ and SGC996 gallbladder cancer cells was 4.47 μM and 5.05 μM, respectively [241], while on human kidney epithelial cells (293 T), dioscin was not toxic even at the higher tested dose (8 μM) [241]. However, Ma et al. reported that dioscin at doses over 10 μM (48 h treatment) inhibited cell proliferation of both gastric cancer (HGC-27, MGC803, and SGC7901) and normal gastric GES-1 cells [242], even if they did not explicitly quantify the entity of this antiproliferative effect.

Questionable data were also found for galangin. Despite galangin’s ability to induce different types of cell death, its active concentration on all the three glioma cell lines tested is quite high (150 µM) [207], and usually high doses are not selectivite towards tumor cells [243,244]. The analysis of galangin effects on normal human astrocytes (NHA) viability showed cytotoxic effects at double the active dose in tumor cells. Indeed, the IC_50_ on NHA after 24 h galangin treatment was >450 μM, whereas in U251, U87MG, and A172 glioma cells the IC_50_ was 221.8, 262.5 and 273.9 μM, respectively [207]. However, 24 h galangin treatment (at concentrations >50 μM) suppressed cell proliferation in NIH3T3 mouse fibroblasts to the same extent as for B16F10 murine melanoma cells. For the latter cell line, the IC_50_ after 24 h treatment was 145 μM, while no data are available for fibroblasts [245].

The cytotoxicity of the pyroptosis inducer osthole was explored on normal cervical fibroblasts and HeLa cervical cancer cells. On HeLa cells, the IC_50_ was 64.9 μM compared to 168 μM on normal cervical fibroblasts (24 h treatment) [246]. Moreover, the IC_50_ on HL-60 after 12 h osthole treatment was 100 μM, compared to 164 μM on PBMCs [247]. Consistently, no significant cytotoxicity was observed in PBMCs up to 72 h osthole treatment at 1.84 μM [248]. Furthermore, osthole treatment for 24 and 48 h at 200 μM did not induce any significant cytotoxic effect on normal ovarian IOSE80 cells [249]. Conversely, almost all A2780 and OV2008 ovarian cancer cells succumbed to osthole treatment at 200 μM for 24 and 48 h [249]. In this regard, the IC_50_ on A2780 and OVCAR3 ovarian cancer cells, i.e., the in vitro cell models where osthole promoted pyroptosis, was 73.6 μM and 75.2 μM, respectively [211]. This means that the active concentrations of osthole are abundantly lower than those toxic for normal ovarian cells.

On the whole, the majority of the studies listed above are encouraging on the, at least a partial, tumour selectivity of non-canonical cell death inducers, but data are far from be conclusive or substantial. One point to consider is that, as for many natural anticancer agents, both activity and selectivity of non-canonical cell death inducers depend on the cell-type and organ targeted. This, together with the lack of extensive studies, does not allow to draw firm conclusions. Thus, since the selective activity of anticancer agents is considered one of the most critical aspects in defining their pharmaco-toxicological profiles, a case-by-case analysis is recommended.

## 6. Conclusions

The ability of cancer cells of evading apoptosis is one of the hallmarks of cancers [250]. Given that anticancer activity of most anticancer drugs currently in use is based on their pro-apoptotic activity [251], it becomes clear how the discovery and characterization of non-apoptotic, also called non-canonical, cell death pathways represent a new promising approach to overcome the challenges of current anticancer therapies. As showed in this review, natural products can definitely suit this role, as promising non-canonical cell death inducers.

All PCD modalities—apoptosis, necroptosis, ferroptosis and pyroptosis—are strictly connected in both molecular and functional terms, and, depending on the cell status or eventually mutations carried by cells, the mode of cell death could switch from one to another [5]. For example, necroptosis occurs when the apoptotic cell death is impaired by caspase-8 inhibition [252]; conversely, within massive inflammasome activation, cells lacking caspase-1 or GSDMD could be unable to trigger pyroptosis but still die by apoptosis thanks to the presence of active caspase-8 [253]. Moreover, activation of apoptotic caspase-3 cleaves GSDME and could trigger both pyroptosis and apoptosis [174,199]. Additionally, RIP3 could activate NLRP3 inflammasome in the absence of MLKL [254] together with the RIP3-MLKL-NLPR3-caspase-1 axis, thus resulting in IL-1β maturation, independently of GSDMD cleavage [255,256]. Hence, all these pieces of evidence show that the different PCDs frequently share the same molecular actors, which could activate different cell death modalities depending on the factors described above. Therefore, we could actually consider all these pathways as many musicians who take part of the same orchestra, and the more musicians play, the marrier is the symphony. In other words, triggering more than one type of PCDs clearly enhances the chances of cancer cells eradication.

Another significant outcome deriving from the concomitant activation of apoptosis and the non-canonical cell deaths is the elicitation of the antitumor immune response, which would allow a switch from a mostly immune-silent or tolerogenic cell death (apoptosis) into an immunogenic one [257,258,259]. For instance, pyroptosis induction commuted the immune-silent cisplatin-mediated apoptosis into immunogenic [260]. Indeed, in different models, GSDME activation promotes tumor suppression by increasing the anticancer properties of tumor-infiltrating natural killer (NK) cells and CD4^+^ and CD8^+^ T lymphocytes, together with an antitumor vaccination effect, triggering both innate and adaptive antitumor immunity [170,171,259]. Similarly, necroptotic cancer cells are, without any doubt, immunogenic. These dying cells promoted antitumor immunity by inducing DC (dendritic cells) maturation, cross-priming and proliferation of CD8^+^ T cells and NK cells in vitro and in vivo and successfully created an antitumor vaccine effect in different tumor mice models [261,262,263,264]. Regarding ferroptosis, many hints have been produced about the interaction between ferroptotic cells and the immune system. For instance, ferroptotic cells release HMGB-1 (high mobility group box 1) [265], while the activation of CD8^+^ T cells synergizes ferroptosis [266]. However, only recently the antitumor immunogenicity of ferroptosis has been validated. Indeed, it has been demonstrated that early, but not late, ferroptotic cells promote the phenotypic maturation of bone-marrow derived dendritic cells (BMDCs) and elicit an antitumor vaccination effect in the well-accepted prophylactic tumor vaccination model of immune competent C57BL/6 J mice [93]. Those results definitely confirm that ferroptosis could promote antitumor immunity. Still, the coexistence of apoptosis and non-canonical cell deaths could be regarded as a new remarkable strategy to neutralize apoptosis resistance and, thanks to the adaptive immune stimulation, lessen the incidence of metastases and relapses.

Nonetheless, this apparently idyllic scenario displays different problems. Induction of necroptosis and pyroptosis is strictly related to the expression of their molecular mediators, which is cancer-type-dependent. For necroptosis, decreased RIP1/RIP3/MLKL expression has been found in AML, melanoma, and breast, colorectal, gastric, ovarian, head and neck squamous cell, and cervical squamous cell carcinomas [96]. Regarding pyroptosis-related mediators, GSDMD expression was found to be decreased in gastric cancer [168], while GSDME expression is low in gastric and skin cancer [168,175]. Thus, the presence or absence of crucial mediators dictates whether cells can go through that specific PCD or not. However, to overcome this limitation and exploit natural compounds’ great potential to induce non-canonical cell death, nanotechnologies can come to the aid. Several nanomaterials demonstrated to counteract the specific pitfalls of every single type of cell death that usually limit their therapeutical use, such as GSDMs silencing for pyroptosis [260] or RIP1/RIP3/MLKL low levels for necroptosis [267], restoring the capability of pursuing that PCD in those resistant models.

Although a huge number of natural compounds has been identified as inducers of non-canonical cell death, only few of them have been deeply characterized for the underpinned molecular networks involved in their antitumor activity. Furthermore, very few studies have investigated the selective activity towards cancer cells together with the drawing of a toxicological profile. This is a critical issue since thee three mentioned non-canonical cell deaths are pro-inflammatory and in some circumstances could promote tumor progression [268,269,270,271,272]. Overall, natural products antitumor potential should be evaluated on a case-by-case basis.

In conclusion, natural products have proven to be interesting and promising non-canonical cell death inducers. However, taking into account all the issues mentioned above, further studies are needed to better characterize their antitumor activity and, especially, to investigate their toxicological profile in order to define their antitumor potential and pave the way for clinical studies.

## Figures and Tables

**Figure 1 cancers-13-00304-f001:**
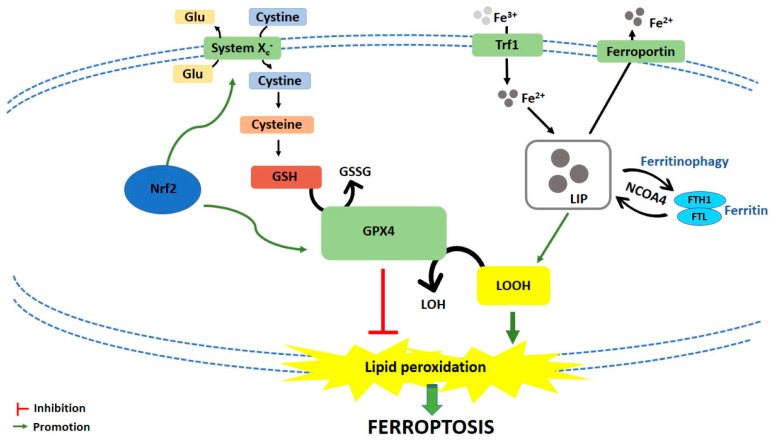
Schematic representation of ferroptotic cell death pathway. Glu: Glutamate; GSH: Glutathione; GSSG: Oxidized glutathione; GPX4: Glutathione peroxidase 4; LOH: Lipid alcohols; LOOH: Lipid hydroperoxides; Nrf2: Nuclear factor (erythroid-derived 2)-like 2; Trf1: Transferrin receptor 1; LIP: Labile iron pool; FTH1: Ferritin heavy chain 1; FTL: Ferritin light chain; NCOA4: Nuclear receptor coactivator.

**Figure 2 cancers-13-00304-f002:**
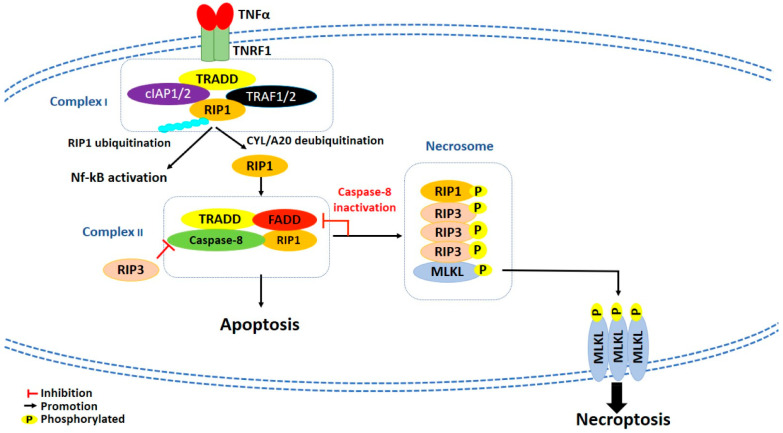
Schematic representation of necroptotic cell death pathway. cIAP1/2: Cellular inhibitors apoptosis proteins; CYL: Cylindromatosis; FADD: Fas-associated death domain; MLKL: Mixed lineage kinase domain-like; Nf-kB: Nuclear factor kappa-light-chain-enhancer of activated B cells; RIP1: Receptor-interacting protein 1; RIP3: Receptor-interacting protein 3; TNFα: Tumor necrosis factor alfa; TNFR1: Tumor necrosis factor receptor 1; TRADD: TNF Receptor-associated death domain; TRAF1/2: TNFR-associated factors 1/2.

**Figure 3 cancers-13-00304-f003:**
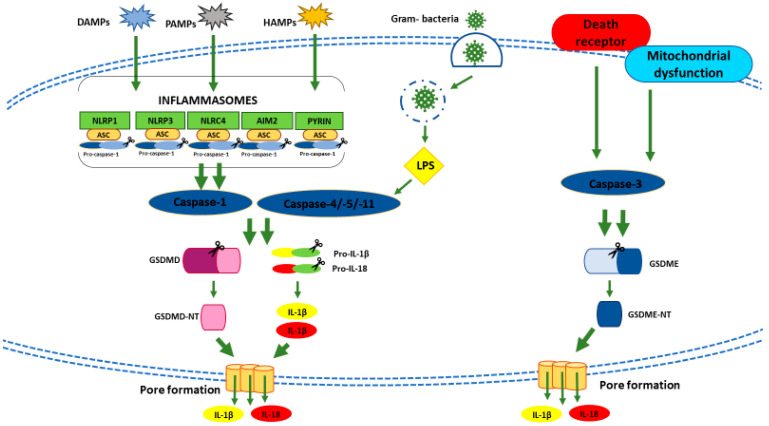
Schematic representation of canonical, non-canonical, and caspase-3-dependent pyroptotic cell death pathway. AIM2: Absent in melanoma 2; ASC: Apoptosis-associated speck-like protein containing a CARD (caspase activation and recruitment domain); DAMPs: Damage-associated molecular patterns; GSDMD: Gasdermin D; GSDMD-NT: N-terminal fragment of GSDMD; GSDME: Gasdermin E; GSDME-NT: N-terminal fragment of GSDME; HAMPs: Homeostasis-altering molecular processes; IL-1β: Interleukin-1 beta; IL-18: Interleukin-18; LPS: Lypopolisaccaride; NLRC4: NLR (nucleotide-binding oligomerization domain (NOD)-like receptor) family CARD domain-containing protein 4; NLRP3: NLR family pyrin domain-containing 3; NLRP1: NLR family pyrin domain-containing 1; PAMPs: Pathogen-associated molecular patterns.

**Table 1 cancers-13-00304-t001:** Natural products as in vitro inducers of ferroptosis.

Compound	Compound Source	Cell Line(s)	Concentrations (Where Specified)	Time (Where Specified)	Ferroptosis Markers	Supplementary Effects	Reference
*Actinia chinensis* (Planch), drug-containing rat serum	*Actinia chinensis* Planch	HGC-27	90, 180 and 360 mg/mL	48 h	↓ Cell proliferation		[25]
24 and 48 h	↓ Cell migration	
180 mg/mL	48 h	↑ ROS	↓ after Ferr-1 treatment
90, 180 and 360 mg/mL	↓ GPX4	
↓ xCT	
Albiziabioside A	*Albizia inundata* Mart.	MCF-7	10 μM	24 h	↑ Cytotoxicity	↑ after Fe^2+^ treatment	[26]
↓ after Ferr-1 treatment
↓ after DFO treatment
↓ after vitamin E treatment
/	↑ ROS	
24 h	↓ GSH/GSSG ratio	
48 h	↓ GPX4 protein expression	
/	↑ MDA	
↑ Lipid peroxides	
Amentoflavone	*Selaginella* spp. and other plants	U251, U373	10 and 20 μM	/	↑ Fe^2+^		[27]
↓ FTH	↑ after ATG7 knockdown
↑ MDA	↓ after FTH overexpression
↑ Lipid ROS	↓ after FTH overexpression
↓ after BafA1 treatment
↓ after ATG7 knockdown
↓ GSH	↓ after FTH overexpression
↓ after BafA1 treatment
↓ after ATG7 knockdown
20 μM	↑ Cell death ratio (%)	↓ after Ferr-1 treatment
↓ after DFO treatment
↓ after FTH overexpression
↓ after BafA1 treatment
↓ after ATG7 knockdown
Ardisiacrispin B	*Ardisia kivuensis* Taton	CCRF-CEM	0.59, 0.93, 2.33, 4.66, 9.32, 18.64 and 37.28 μM	24 h	↑ Cytotoxicity	↓ after Ferr-1 treatment	[28]
↓ after DFO treatment
0.3, 0.6, 1.2 and 2.4 μM	↑ ROS	
Aridanin	*Tetrapleura tetraptera* (Schum. & Thonn) Taub.	CCRF-CEM	1, 2, 4, 8, 15, 30 and 61 μM	24 h	↓ Cell viability	↓ after Ferr-1 treatment	[29]
↓ after DFO treatment
Artenimol (artemisinin semi-syntethic derivative)	*Artemisia annua* L.	CCRF-CEM	0.01, 0.1, 1, 10 and 100 μM	/	↓ Cell viability	↓ after Ferr-1 treatment	[30]
↓ after DFO treatment
Artesunate (artemisin semi-synthetic derivative)	*Artemisia annua* L.	DAUDI, CA-46	4 and 20 μM	48 h	↓ Cell viability	↑ after DFO treatment	[31]
↑ after Ferr-1 treatment
↑ after Lip-1 treatment
↑ after down-regulation of CHAC1 expression
5, 10 and 20 μM	24 and 48 h	↑ ROS	
↑ Lipid peroxidation	↓ after down-regulation of CHAC-1 expression
5, 10 and 20 μM	24 h	↑ CHAC1, ↑ ATF4, ↑ CHOP protein expression	
MT-2	50 μM	24 h	↑ ROS		[32]
0.4, 2 and 10 μM	↑ Cytotoxicity	↓ after DFO treatment
2 and 10 μM	↓ after Ferr-1 treatment
HUT-102	50 μM	24 h	↑ ROS	↓ after NAC treatment
2 and 10 μM	↑ Cytotoxicity	↓ after DFO treatment
10 and 50 μM	↓ after Ferr-1 treatment
HN9	50 μM	72 h	↓ Cell viability	↓ after HTF treatment	[33]
↑ after DFO treatment
↑ after Trolox treatment
2.5 and 5 μM	↑ after Keap1 knockdown
↑ after Nrf2 knockdown
50 μM	24 h	↑ ROS	↓ after Ferr-1 treatment
↓ after Trolox treatment
↑ Lipid ROS	↓ after Ferr-1 treatment
↓ after Trolox treatment
HN9, HN9-cisR	10, 25 and 50 μM	24 h	↑ Nrf2 protein expression	
↓ xCT, ↓ RAD51, ↓ Keap1 protein expression	
HN9-cisR, HN3-cisR, HN4-cisR	10, 25 and 50 μM	24 h	↑ Nrf2, ↑ HO-1, ↑ NQO1 protein expression	
↓ Keap1 protein expression	
50 μM	↑ Nrf2, ↑ HO-1, ↑ NQO1 mRNA levels	
HN3-cisR	25 and 50 μM	24 h	↓ GSH	↓ after trigonellin treatment
↑ after Trolox treatment
↑ after Nrf2 knockdown
↑ ROS	↓ after Trolox treatment
↑ after Nrf2 knockdown
↑ Lipid ROS	↓ after trigonellin treatment
↓ Cell viability	↓ after Nrf2 knockdown
↓ after HO-1 knockdown
↑ after Trolox treatment
PaTU8988, AsPC-1	20 μM	24 h	↓ Cell viability	↑ after Ferr-1 treatment	[34]
↑ after GRP78 overexpression
↓ after GRP78 knockdown
↑ MDA	↓ after DFO treatment
↓ after Ferr-1 treatment
↓ after GRP78 overexpression
↑ after GRP78 knockdown
↑ Lipid peroxidation	↓ after Ferr-1 treatment
10, 20 and 40 μM	↑ GRP78 mRNA levels	
↑ GRP78 protein expression	
HEY1	25 and 50 μM	48 h	↑ Cell death	↓ after Ferr-1 treatment	[35]
↓ after DFO treatment
↑ after HT treatment
HEY2	100 μM	↑ Cell death	↓ after Ferr-1 treatment
HEY2, SKOV3	50 and 100 μM	↑ Cell death	↓ after DFO treatment
↑ after HT treatment
HEY1, HEY2, SKOV-3	10, 25, 50 and 100 μM	24 h	↑ ROS	↓ after GSH treatment
HEY1, HEY2, SKOV-3, OVCAR8, TOV-112D, TOV-21G	25, 50 and 100 μM	48 h	↑ Cell death	↓ after GSH treatment
Panc-1	50 μM	24 h	↑ ROS	↓ after Trolox treatment	[36]
↓ Colony formation	↑ after DFO treatment
↑ after Trolox treatment
↑ after Ferr-1 treatment
↓ after HTF treatment
↑ HO-1 protein expression	
↑ Lipid peroxidation	↓ after Trolox treatment
↓ after Ferr-1 treatment
Panc-1, COLO-357	48 h	↑ Cell death	↓ after Ferr-1 treatment
BxPC-3, Panc-1	24 and 48 h	↑ Cell death	↓ after DFO treatment
BxPC-3, Panc-1, AsPC-1	↑ Cell death	↑ after HTF treatment
*Betula etnensis* Raf. methanolic extract	*Betula etnensis* Raf.	CaCo2	5, 50, 250 and 500 μg/mL	72 h	↓ Cell viability		[37]
5, 50 and 250 μg/mL	↑ LDH release	
↑ ROS	
↑ LOOH	
↓ RSH	
5 and 50 μg/mL	↓ HO-1 levels	
250 μg/mL	↑ HO-1 levels	
D13 (albiziabioside A derivative)	*Albizia inundata* Mart.	HCT116	0.31, 1.25 and 5 μM	/	↑ Cytotoxicity	↓ after Ferr-1 treatment	[38]
↓ after DFO treatment
↑ after Fe^2+^ treatment
↑ after Fe^3+^ treatment
48 h	↓ GPX4 protein expression	
/	↑ MDA	
Dihydroartemisinin (artemisin semi-synthetic derivative)	*Artemisia annua* L.	THP-1	5, 10 and 15 μM	12 h	↓ Cell viability		[39]
↑ ROS	
HL-60	5, 10 and 15 μM	12 h	↓ Cell viability	↑ after Ferr-1 treatment
↑ after DFO treatment
↑ after NAC treatment
↑ after BafA1 treatment
↑ after 3-MA treatment
↑ after ATG7 knockdown
↑ after FTH overexpression
↑ after ISCU overexpression
↑ Lipid ROS	↓ after ATG7 knockdown
↓ after FTH overexpression
↓ GSH	↑ after ISCU overexpression
↑ ROS	↓ after DFO treatment
↓ after NAC treatment
↓ after ISCU overexpression
↑ IRP2 protein expression	
↓ FTH, ↓ GPX4 protein expression	↑ after ISCU overexpression
↑ after BafA1 treatment
G0101, G0107	10, 20, 40, 80 and 160 μM	24 h	↑ ROS		[40]
20, 40, 80 and 160 μM	↑ Lipid ROS	
↑ MDA	
↓ GSH	
↑ GSSG	
↑ Cell death	↓ after DFO treatment
↓ after Ferr-1 treatment
↓ after Lip-1 treatment
U251U373	5, 10, 20 and 40 μM20, 40, 80 and 160 μM	24 h	↓ GSH	

U251U373	2.5, 5, 10, 20 and 40 μM10, 20, 40, 80 and 160 μM	24 and 48 h	↑ ROS	↓ after DFO treatment
↑ after PERKi treatment
↑ after ATF4 siRNA treatment
↑ after HSPA5 siRNA treatment
U251U373	2.5, 5, 10 and 20 μM10, 20, 40, 80 and 160 μM	3, 6, 12, 24 and 48 h	↑ Lipid ROS	↑ after ATF4 siRNA treatment
↑ after HSPA5 siRNA treatment
U251U373	5, 10, 20 and 40 μM80 μM	3, 6, 12, 24 and 48 h	↑ MDA	↑ after PERKi treatment
↑ after ATF4 siRNA treatment
↑ after HSPA5 siRNA treatment
U251U373	10, 20 and 40 μM40, 80 and 160 μM	48 h	↑ Cell death	↓ after DFO treatment
↓ after Ferr-1 treatment
↓ after Lip-1 treatment
↑ after PERKi treatment
↑ after ATF4 siRNA treatment
↑ after HSA5 siRNA treatment
Dihydroisotanshinone I	*Salvia miltiorrhiza* Bunge	MCF-7	5 and 10 μM	24 h	↓ GPX4 activity		[41]
MCF-7, MDA-MB231	10 μM	↓ GPX4 protein expression	
5 and 10 μM	↑ MDA	
10 μM	↓ GSH/GSSG ratio	
Epunctanone	*Garcinia epunctata* Stapf.	CCRF-CEM	1.04, 1.66, 4.14, 8.28, 16.56, 33.11 and 66.23 μM	24 h	↑ Cytotoxicity	↓ after Ferr-1 treatment	[42]
↓ after DFO treatment
2.95, 5.91, 11.81 and 23.63 μM	↑ ROS	
Erianin	*Dendrobium chrysotoxum* Lindl	H460, H1299	50 and 100 nM	24 h	↑ Cell death	↓ after NAC treatment	[43]
↓ after Ferr-1 treatment
↓ after Lip-1 treatment
↓ after GSH treatment
50 and 100 nM	/	↑ ROS	
↓ GSH	
12.5, 25, 50 and 100 nM	↑ MDA	
↑ HO-1, ↑ transferrin protein expression	
↓ GPX4, ↓ CHAC2, ↓ SLC40A1, ↓ SLC7A11 protein expression	
5, 10 and 25 μM	24 h	↑ Ca^2+^ levels	
↑ Calmodulin protein expression	
Ferroptocide (pleuromutilin semi-syntetic derivative)	*Pleurotus passeckerianus; Drosophila subatrata; Clitopilus scyphoides,* and others spp.	ES-2	5, 10 and 25 μM	1 h	↑ ROS	↓ after DFO treatment	[44]
10 and 25 μM	↑ Mitochondrial ROS	
10 μM	↑ Lipid ROS	↓ after DFO treatment
5, 10 and 25 μM	14 h	↑ Cell death	↓ after Ferr-1 treatment
↓ after DFO treatment
↓ after Trolox treatment
HCT116	5, 10 and 25 μM	10, 24 and 48 h	↑ Cell death	↓ after DFO treatment
↓ after Trolox treatment
↓ after NAC-1 treatment
↑ after TXN knockdown
↓ after Ferr-1 treatment
5, 10 and 25 μM	1.5 and 72 h	↑ ROS	↑ after TXN knockdown
10 μM	2 and 72 h	↑ Lipid ROS	↑ after TXN knockdown
↓ after DFO treatment
4T1	5, 10 and 25 μM	18 h	↑ Cell death	↓ after DFO treatment
↓ after Ferr-1 treatment
10 μM	2 h	↑ Lipid ROS	↓ after Ferr-1 treatment
↓ after DFO treatment
HT-29	5, 10 and 25 μM	12 h	↑ Cell death	↓ after DFO treatment
↓ after Trolox treatment
↓ after NAC treatment
↓ after Ferr-1 treatment
Gallic Acid	Natural polyhydroxy phenolic compound, found in various foods	HeLa	50 μg/mL	12 h	↑ Lipid peroxidation		[45]
HeLa, H446, SHSY-5Y	36 h	↑ Cell death	↓ after DFO treatment
A375, MDA-MB-231	10, 25, 50, 100 and 200 μg/mL	24 h	↓ Cell viability		[46]
MDA-MB-231	25 μg/mL	↑ ROS	
A375	50 μg/mL	
MDA-MB-231	/	/	↓ GPX4 activity	
A375, MDA-MB-231	/	/	↑ MDA	
Physcion 8-O-β-glucopyranoside	*Rumex japonicus* Houtt.	MGC-803, MKN-45	10, 20, 30, 40 and 50 μM	24, 48, 72 and 96 h	↓ Cell viability	↑ after Ferr-1 treatment	[47]
↑ after GPNA treatment
↑ after 968 treatment
↓ after GLS2 knockdown
/	/	↓ Cell proliferation	↑ after miR-103a-3p overexpression
/	24 h	↓ Cell invasion	↑ after miR-103a-3p overexpression
↓ Cell migration	↑ after miR-103a-3p overexpression
/	/	↑ Lipid ROS	↓ after GPNA treatment
↓ after 968 treatment
↓ after GLS2 knockdown
↓ after miR-103a-3p overexpression
↑ MDA	↓ after GPNA treatment
↓ after 968 treatment
↓ after GLS2 knockdown
↓ after miR-103a-3p overexpression
↑ Fe^2+^	↓ after GPNA treatment
↓ after 968 treatment
↓ after GLS2 knockdown
↓ after miR-103a-3p overexpression
↓ miR-103a-3p expression	
↑ GLS2 protein levels	↓ after miR-103-3p transfection
Piperlongumine	*Piper Longum* L.	Panc-1	4, 6, 8, 10, 12 and 14 μM	16 h	↓ Cell viability	↑ after NAC treatment	[48]
↑ after Ferr-1 treatment
↑ after Lip-1 treatment
↑ after DFO treatment
MIAPaCa-2	10 μM	16 h	↓ Cell viability	↑ after CPX treatment
↑ after PD146176 treatment
4 h	↓ GSH	
Progenin III	*Raphia vinifera* P. Beauv	CCRF-CEM	2, 3, 7, 14 and 55 μM	24 h	↓ Cell viability	↑ after Ferr-1 treatment	[49]
↑ after DFO treatment
1.59 and 3.18 μM	↑ ROS	
Ruscogenin	*Ruscus aculeatus* L. *Radix Ophiopogon japonicas* (Thunb.) Ker Gawl.	BxPC-3, SW1990	7 μM	/	↑ Cell death	↑ after FAC treatment	[50]
↓ after DFO treatment
/	6 h	↑ Cell death	↓ after transferrin knockdown
↓ after ferroportin overexpression
3 and 7 μM	12 and 24 h	↑ Fe^2+^	↓ after DFO treatment
1, 2, 4, 6 and 24 h	↑ ROS	↓ after DFO treatment
6 and 12 μM	24 h	↑ Transferrin	
↓ Ferroportin	
Solasonine	*Solanum melongena* L.	HepG2	15 ng/mL	24 h	↑ Cell death	↓ after Ferr-1 treatment	[51]
↓ after DFO treatment
↑ Lipid ROS	↓ after Ferr-1 treatment
↓ after DFO treatment
↓ GSS, ↓ GPX4 mRNA levels	
↓ GSS, ↓ GPX4 protein expression	
Typhaneoside	*Pollen Typhae*	Kas-1, HL-60, NB4	40 μM	24 h	↓ Cell viability	↑ after Ferr-1 treatment	[52]
↑ after DFO treatment
↑ after 3-MA treatment
↑ after BafA1 treatment
↑ after Z-VAD-FMK treatment
↑ after rapamycin treatment
↑ after ATG7 knockdown
20, 30 and 40 μM	↑ ROS	↓ after DFO treatment
↓ after NAC treatment
↓ GSH	
↑ Lipid ROS	↓ after ATG7 knockdown
↓ after BafA1 treatment
↓ GPX4, ↓ FTH mRNA levels	
↑ IRP2 mRNA levels	
Ungeremine	*Crinum zeylanicum* L.	CCRF-CEM	2.37, 3.76, 9.40, 18.79, 37.58, 75.17 and 150.33 μM	24 h	↓ Cell proliferation	↑ after Ferr-1 treatment	[53]
↑ after DFO treatment
1.22, 2.45, 4.89 and 9.78 μM	↑ ROS	
Whitaferin A	*Withania somnifera* (L.) Dunal	IMR-32	/	/	↑ ROS		[54]
1 and 10 μM	2, 4, 8, 12 and 24 h	↓ GPX4 expression	
10 μM	3 and 5 h	↓ GPX4 activity	
/	/	↑ Lipid peroxidation	↓ after DFO treatment
1 μM	4, 8 and 12 h	↑ Fe^2+^	↑ after hemin treatment
1, 2, 4, 8, 12 and 24 h	↑ HO-1, ↑ Keap1, ↑ Nrf2 protein expression	
6, 8, 12 and 16 h	↑ Cell death	↓ after GPX4 overexpression
↓ after ZnPP treatment
↓ HO-1 knockdown
↑ after hemin treatment
IMR-32, SK-N-SH	1 and 10 μM	6, 8, 12 and 16 h	↑ Cell death	↓ after Ferr-1 treatment
↓ after CPX treatment
↓ after α-tocopherol treatment
↓ after UOI26 treatment
↓ after Flt3 inhibitor treatment
1 μM	/	Nrf2 pathway activation	
/	↑ FTH1, ↑ HO-1 gene expression	
1, 2, 4, 8, 12 and 24 h	↑ FTH1, ↑ HO-1 mRNA levels	
WA-NPs	*Withania somnifera* L. Dunal	IMR-32	1 and 10 μM	8, 10, 12, 16, 20 and 24 h	↑ Cell death		[54]

*Abbreviations*: ↑: Increase; ↓: Decrease; 3-MA:3-methyladenine; 968: Compound 968, GLS2 inhibitor;; ATF4: Activating transcription factor 4; ATG7: Autophagy related 7; BafA1: Bafilomycin 1; CHAC1: Glutathione-specific Gamma-glutamylcyclotransferase 1; CHOP: CCAAT/enhancer-binding protein homologous protein; CPX: Ciclopirox, intracellular iron chelator; DFO: Deferoxamine; FAC: Ferric ammonium citrate; Fe^2+^: Ferrous ion; Fe^3+^: Ferric ion; Ferr-1: Ferrostatin-1; Flt3: Receptor tyrosine kinase fms-like tyrosine kinase 3; FTH: Ferritin heavy chain; FTH1: Ferritin heavy chain 1; GLS2: Glutaminase 2; GPNA: Glutamine transporter inhibitor; GPX4: Glutathione peroxidase IV; GSH: Glutathione; GSS: Glutathione synthetase; GSSG: Oxidized glutathione; HO-1: Heme oxygenase 1; HN3-cisR: Cisplatin-resistant HN3 cells; HN4-cisR: Cisplatin-resistant HN4 cells; HN9-cisR: Cisplatin-resistant HN9 cells; HSPA5: Heat shock protein family A (Hsp70) member 5; HTF: Holo-transferrin; IRP2: Iron regulator protein 2; ISCU: Iron-sulfur cluster assembly enzyme; Keap1: Kelch-like ECH-associated protein 1; Lip-1: Liproxstatin-1; MDA: Malondialdehyde; NAC: N-acetylcysteine; NQO1: NAD(P)H quinone dehydrogenase 1; Nrf2: Nuclear factor erythroid 2–related factor 2; PD146176: Lypoxygenase inhibitor; PERKi: PERK inhibitor I (GSK2606414); ROS: Reactive oxygen species; RSH: Thiols; Spp.: Species; TXN: Thioredoxin; WA-NPs: Whitaferin A nanoparticles; xCT: Cystine/glutamate antiporter; ZnPP: Zinc protoporphyrin, HO-1 inhibitor; Z-VAD-FMK: Pan-caspase inhibitor.

**Table 2 cancers-13-00304-t002:** Natural compounds as in vitro inducers of necroptosis.

Compound	Compound Source	Cell Line(s)	Concentrations (Where Specified)	Time (Where Specified)	Necroptosis Markers	Supplementary Effects	Reference
2-methoxy-6-acetyl-7-methyljuglone	*Polygonum cuspidatum* Sieb. et Zucc.	A549	5 μM	24 h	Intact nuclear envelope		[110]
Mitochondrial swelling	
Loss of mitochondrial matrix	
Cytoplasm vacuolization	
2.5, 5 and 7.5 μM	↑ LDH release	
2.5, 5 and 7.5 μM	16 h	No caspase-3/-7 activation	
5 and 7.5 μM	24 h	↓ Cell viability	↑ after Nec-1 treatment
↑ after SP600125 treatment
↑ after JNK knockdown
↑ after NAC treatment
↑ after GSH treatment
↑ after CAT treatment
↑ after DTT treatment
↑ after Hgb treatment
↑ after iNOS knockdown
7.5 μM	1, 3, 6 and 16 h	↑ p-JNK protein expression	↓ after SP600125 treatment
↓ after CAT treatment
↑ p-P38 protein expression	↓ after SB203580 treatment
/	1, 2 and 4 h	↑ ROS	↓ after NAC treatment
↓ after GSH treatment
7.5 μM	1 h	↓ GSH/GSSG ratio	↑ after GSH treatment
	1, 2 and 4 h	↑ H_2_O_2_	
2 and 4 h	↑ NO	↓ after SP600125 treatment
↑ after JNK knockdown
↑ HROS	
↑ O_2_^−^	
7.5 μM	1, 3, 6, 16 and 24 h	↑ iNOS protein expression	
5 and 7.5 μM	6 h	↑ NOS activity	↓ after L-NMMA treatment
2.5, 5 and 7.5 μM	24 h	↓ p-IκBα, ↓ NF-kB protein expression	
10 μM	1, 2 and 4 h	↑ Lipid peroxidation	
A549	7.5 μM	8 h	Swollen mitochondria		[111]
Damaged cell membrane	
A549H1299	7.5 μM2.5 μM	1, 2, 4 and 8 h	No caspase-3/-8/-9 cleavage	

No PARP cleavage	

↑ p-RIP1, ↑ p-RIP3, ↑ p-MLKL protein expression	
A549H1299	/	8 h4 h	RIP1-RIP3 interaction	↓ after Nec-1s treatment
A549H1299	7.5 μM2.5 μM	1 h	↑ ROS	↓ after Nec-1s treatment
A549H1299	7.5 μM2.5 μM	4 h2 and 4 h	↑ Ca^2+^	↓ after Nec-1s treatment
↓ after SP600125 treatment
↓ after BAPTA-AM treatment
↓ after CAT treatment
A549H1299	7.5 μM2.5 μM	4 h	↑ JNK1/2, ↑ p-JNK1/2 protein expression	↓ after Nec-1s treatment
↓ after SP600125 treatment
↓ after BAPTA-AM treatment
↓ after CAT treatment
A549H1299	7.5 μM2.5 μM	1 and 2 h	Lysosomal membrane permeabilization	↓ after Nec-1s treatment
↓ after SP600125 treatment
↓ after BAPTA-AM treatment
↓ after CAT treatment
↓ after Hgb treatment
A549	7.5 μM	4 h	↑ Mitochondrial ROS	↓ after Nec-1s treatment
↓ after SP600125 treatment
↓ after BAPTA-AM treatment
↓ after CAT treatment
↓ after MnSOD overexpression
H1299	2.5 μM	4 h	↑ Mitochondrial ROS	↓ after Nec-1s treatment
↓ after SP600125 treatment
↓ after BAPTA-AM treatment
↓ after CAT treatment
A549H1299	7.5 μM2.5 μM	4 h	↓ ΔΨm	↑ after Nec-1s treatment
↑ after SP600125 treatment
↑ after BAPTA-AM treatment
↑ after CAT treatment
H1299	2.5 μM	6 h	↓ Cell viability	↑ after Nec-1s treatment
↑ after BAPTA-AM treatment
↑ after K45A treatment
↑ after DI60N treatment
A549	7.5 μM	8 h	↓ Cell viability	↑ after Nec-1s treatment
↑ after BAPTA-AM treatment
↑ after K45A treatment
↑ after DI60N treatment
↑ after MnSOD overexpression
A549/Cis	10 μM	72h	↓ Cell viability	↑ after Nec-1s treatment
Extensive vacuolation	
Damaged and swollen mitochondria	
Intact cell nuclei	
U87, U251	5, 7.5 and 10 μM	8 h	↑ PI positive cells		[112]
/	8 h	↑ LDH release	
5, 7.5 and 10 μM	/	No caspase-3/-7 activation	
/	1, 2 and 4 h	↑ O_2_^−^ generation	↓ after NAC treatment
↓ after DIC treatment
2, 4 and 8 h	↑ Cytosolic Ca^2+^ accumulation	↓ after BAPTA-AM treatment
↓ after NAC treatment
0.5, 1, 2, 4 and 8 h	↑ p-CaMKII protein expression	
0.5, 1 and 2 h	↑ p-JNK1/2 protein expression	↓ after BAPTA-AM treatment
↓ after NAC treatment
8 h	↓ Cell viability	↑ after NAC treatment
↑ after GSH treatment
↑ after CAT treatment
↑ after DTT treatment
↑ after BAPTA-AM treatment
↑ after KN93 treatment
↑ after SP600125 treatment
↑ after DIC treatment
↑ after NQO1 knockdown
U251	/	4 h	Intracellular bubbles	
1 and 2 h	Mitochondrial fragmentation	
4 h	↓ ΔΨm	↑ after SP600125 treatment
↑ after BAPTA-AM treatment
↑ Mitochondrial O_2_^−^ generation	↓ after DIC treatment
↓ after CAT treatment
↓ after SP600125 treatment
↓ after BAPTA-AM
**11-methoxytabersonine**	*Tabernaemontana bovina* Lour.	A549, H157	1, 2.5 and 5 μM	48 h	No caspase-3 activation		[113]
2.5 μM	24 and 48 h	↑ LDH release	↑ after 3-MA treatment
↑ after CQ treatment
2.5 μM	16 and 24 h	RIP1-RIP3 interaction	↑ after 3-MA treatment
2.5, 5, 7.5 and10 μM	48 h	↓ Cell viability	↑ after Nec-1 treatment
↑ after 3-MA treatment
↑ after CQ treatment
***Acridocarpus orientalis* dichloromethane fraction**	*Acridocarpus orientalis* A. Juss	HeLa	250 μg/mL	/	No caspase-3/7-8-9 activation		[114]
24 h	↑ Cell death	↓ after Nec-1 treatment
↓ after Nec-1 + Z-VAD-FMK treatment
***Acridocarpus orientalis* n-butanol fraction**	*Acridocarpus orientalis* A. Juss	HeLa	125 μg/mL	24 h	↑ Cell death	↓ after Nec-1 treatment	[114]
↓ after Nec-1 + Z-VAD-FMK treatment
**Arctigenin**	*Arctium lappa* L., *Saussurea heteromalla*	PC-3, PC-3AcT	20 μM	24 and 48 h	↓ Cell viability	↑ after Nec-1 treatment	[115]
↑ after ATP treatment
20 and 40 μM	48 h	↑ p-RIP3, ↑ p-MLKL protein expression	↓ after Nec-1 treatment
↓ after ATP treatment
↓ after CCN1 knockdown
5, 10, 20 and 40 μM	↑ CCN1 protein expression	↓ after NAC treatment
↓ p-Akt protein expression	↑ after NAC treatment
PC-3, PC-3AcT-cells derived spheroids	20 μM	48 h	↓ Spheroid growth and viability	↓ after NAC treatment
↑ CCN1, ↑ p-RIP3, ↑ p-MLKL protein expression	
RPMI-2650	5 μM	24, 48 and 72 h	↓ Cell viability	↑ after Nec-1 treatment	[116]
↑ after ATP treatment
48 h	↑ Necrotic cells	↓ after NAC treatment
↑ ROS	↓ after NAC treatment
↓ ΔΨm	↑ after NAC treatment
↓ ATP levels	↓ after Nec-1 treatment
/	↑ RIP3, ↑ p-RIP3 protein expression	
↑ MLKL, ↑ p-MLKL protein expression	
↑ p-ATM protein expression	
↑ p-ATR protein expression	
↑ p-CHK1/2 protein expression	
**Aridanin**	*Tetrapleura tetraptera* (Schum. & Thonn) Taub.	CCRF-CEM	1, 2, 4, 8, 15, 30, and 61 μM	24 h	↓ Cell viability	↑ after Nec-1 treatment	[29]
3.18 and 6.36 μM	↑ PI positive cells	
**Artesunate (artemisin semi-synthetic derivative)**	*Artemisia annua* L.	Human Primary schwannoma cells	200 μM	24 h	↓ Cell viability	↑ after Nec-1 treatment	[117]
↓ after CQ treatment (↑ after Nec-1 treatment)
100 μM	20 h	↑ p-MLKL protein expression	
RT4	25 and 50 μM	20 h	↓ Cell viability	↑ after Nec-1 treatment
↑ after RIP1 knockdown
25 and 50 μM	↑ p-MLKL protein expression	
10, 25, 50 and 100 μM	↑ RIP1 protein expression	
Hela, COLO-205	50 μM	20 h	↑ p-MLKL protein expression	
**Berberine**	Huang Lian Chinese herb (*Coptis chinesis* Franch) and *Hydrastis Canadensis* L.	OVCAR3	100 μM	24 h	Extensive vacuolation		[118]
Rupture of plasma membrane	
OVCAR3, POCCLs	↑ RIP3, ↑ MLKL mRNA levels	
↑ RIP3, ↑ MLKL protein expression	
↑ p-RIP3, ↑ p-MLKL protein expression	
DB, RAMOS	6.25, 12.5, 25 and 50 μM	/	↑ Growth inhibition		[119]
DB	30 μM	48 h	Formation of RIP1/RIP3/MLKL complex	
RAMOS	20 μM	
DB	30 μM	48 h	Swollen mitochondria	
Intact cell nuclei	
12 and 24 h	↓ PCYT1A mRNA levels	
24 h	↑ Degradation of PCYT1A mRNA	
/	↓ PCYT1A protein expression	
**Celastrol**	*Tripterygium wilfordii* Hook. f.	HGC-27, AGS	0.25, 0.5, 1 and 2 μM	24 h	↓ Cell viability	↑ after RIP3 knockdown	[120]
↑ after Nec-1 treatment
↑ after Nec-1 + Z-VAD-FMK treatment
↑ after BGN overexpression
↓ after BGN knockdown
↑ after NSA treatment
↑ after Nec-1 treatment+ BGN overexpression
0.5 μM	↑ PI positive cells	↓ after BGN overexpression
0.25, 0.5, 1 and 2 μM	↑ RIP1, ↑ RIP3 protein expression	
↑ p-RIP1, ↑ p-RIP3 protein expression	↓ after BGN overexpression
↓ after Nec-1 treatment
↓ BGN protein expression	
0.5 μM	Cell rounding and shrinkage	↓ after BGN overexpression
↑ MLKL protein expression	
↑ p-MLKL protein expression	↓ after BGN overexpression
↑ MLKL translocation to plasma membrane	↓ after BGN overexpression
↓ TNF-α secretion	↑ after BGN overexpression
↓ IL-8 secretion	↑ after BGN overexpression
**Columbianadin**	*Angelica decursiva* Fr. Et Sav	HCT116	50 μM	48 h	↑ RIP1, ↑ RIP3 protein expression		[121]
↓ Caspase-8 cleavage	
↑ ROS	↓ after CAT treatment
↓ CAT protein expression	
↓ SOD-1/2 protein expression	
**Deoxypodophyllotoxin**	*Pulsatilla koreana* (Yabe ex Nakai) Nakai ex T. Mori	NCI-H460	30 nM	24 h	Rupture of plasma membrane	↓ after Nec-1 treatment	[122]
Cytoplasmatic vacuolation	↓ after Nec-1 treatment
Mitochondria swelling	↓ after Nec-1 treatment
Cytoskeletal degradation	↓ after Nec-1 treatment
Dilation of endoplasmic reticulum elements	↓ after Nec-1 treatment
↑ PI penetration	↓ after Nec-1 treatment
↓ ΔΨm	↑ after Nec-1 treatment
**Emodin**	*Rheum palmatum* L.	U251	/	12 h	↑ LDH release	↓ after Nec-1 treatment	[123]
↓ after GSK-872 treatment
10, 20 and 40 μM	12 h	↑ RIP1 protein expression	↓ after Nec-1 treatment
↑ RIP3 protein expression	↓ after GSK-872 treatment
↑ TNF-α protein expression	
**Gomisin J**	*Schisandra chinensis* (Turcz.) Baill.	MCF-7, MDA-MB-231	30 μg/mL	72 h	↓ Cell viability		[124]
↑ Extracellular CypA protein expression	
**Jujuboside B**	*Zizyphus jujube* Mill var. spinosa (Bunge) Hu ex H. F. Chow	U937	40, 80 and 120 μM	24 h	↑ RIP1, ↑ p-RIP1 protein expression	↓ after Nec-1 treatment	[125]
↑ RIP3, ↑ p-RIP3 protein expression	↓ after Nec-1 treatment
↑ MLKL, ↑ p-MLKL protein expression	↓ after Nec-1 treatment
80 μM	↓ Cell viability	↓ after Nec-1 treatment
↓ Colony formation	↓ after Nec-1 treatment
**Matrine**	*Sophora flavescens* Aiton	Mz-ChA-1, QBC939	1.5 mg/mL	24 and 48 h	Extensive organelle and cell swelling		[126]
Cytoplasmatic vacuolation	
Loss of membrane integrity	
No alterations of nuclei morphology	
48 h	↑ PI positive cells	↓ after Nec-1 treatment
↓ after RIP3 knockdown
↓ after NSA treatment
↓ after NAC treatment
3, 6, 9 and 12 h	↑ RIP3 protein expression	
2 h	↑ MLKL membrane translocation	↓ after Nec-1 treatment
0.25, 0.5, 1, 1.5 and 2 mg/mL	24 h	↑ ROS	↓ after Nec-1 treatment
↓ after NSA treatment
**Neoalbacol**	*Albatrellus confluens*	C666-1	40 μM	/	Cell swelling		[127]
24 h	↓ Cell viability	↑ after Nec-1 treatment
↑ after Akt overexpression
↑ after Nec-1 + 3-MA treatment
↓ after SP600125 treatment
20, 30 and 40 μM	/	RIP1-RIP3 interaction	↓ after Nec-1 treatment
8 h	↓ p-Akt protein expression	↑ after Akt overexpression
↓ p-TSC2, ↓ p-mTOR, ↓ p-p70S6K1 protein expression	
/	/	↓ TNF-α, ↓ EGF, ↓ IL-6 protein expression	
40 μM	/	↓ GLUT1/4 mRNA levels	
↓ HK2 mRNA levels	↑ after Akt overexpression
20, 30 and 40 μM	↓ HK2 protein expression	↑ after Akt overexpression
↓ GLUT1/4 protein expression	
40 μM	2, 4, 6, 8 and 10 h	↓ ATP levels	↑ after Akt overexpression
↓ Glucose concentration	↑ after Akt overexpression
/	↑ p-JNK protein expression	↓ after Nec-1 treatment
HK1	40 μM	24 h	↓ Cell viability	↑ after Nec-1 treatment
↑ after Nec-1 + 3-MA treatment
↓ after SP600125 treatment
20, 30 and 40 μM	8 h	↓ p-Akt, ↓ p-TSC2, ↓ p-mTOR, ↓p-p70S6K1 protein expression	
40 μM	/	↓ GLUT1/4, ↓ HK2 mRNA levels	
**Ophiopogonin D’**	*Ophiopogon japonicus* (Thunb.) Ker Gawl	LNCaP	2.5 and 5 μM	24 h	↑ Necrotic cells	↓ after NSA treatment	[128]
↓ after Nec-1 treatment
↓ after Nec-1 + NSA treatment
5 μM	↓ Cell viability	↑ after Nec-1 treatment
↑ after Nec-1 + NSA treatment
2.5 and 5 μM	6 h	↑ RIP3 protein expression	
↑ MLKL, ↑ p-MLKL protein expression	
5 μM	↑ RIP1 protein expression	
↑ Caspase-8, ↑ cleaved caspase-8 protein expression	
RIP3-MLKL interaction	
↑ FasL protein expression	
↑ Soluble FasL protein expression	↓ after Nec-1 treatment
↓ Fas protein expression	
↑ FADD protein expression	
↓ Bim protein expression	
↓ AR, ↓ PSA protein expression	↓ after Nec-1 treatment
**Pristimerin**	Various plant spp. of *Celastraceae* and *Hippocrateaceae* families	C6U251	2.5 μM4.5 μM	6 h	Loss of membrane integrity		[129]
Intact nuclear membrane	
Swollen mitochondria	
↓ Cell viability	↑ after AIF knockdown
↑ after SP600125 treatment
↑ PI positive cells	↓ after AIF knockdown
↓ after SP600125 treatment
↓ after JNK knockdown
1.5 and 6 h	↓ ΔΨm	↑ after SP600125 treatment
↑ after JNK knockdown
1.5, 3 and 6 h	No caspase-3 activation	
1.5, 3, 6 and 12 h	↑ AIF nuclear translocation	↓ after SP600125 treatment
↓ after JNK knockdown
1.5 and 6 h	↑ JNK protein expression	
↑ p-JNK protein expression	↓ after SP600125 treatment
↓ after JNK knockdown
**Progenin III**	*Raphia vinifera* P. Beauv	CCRF-CEM	2, 3, 7, 14 and 55 μM	24 h	↓ Cell proliferation	↑ after Nec-1 treatment	[49]
1.59 and 3.18 μM	↑ ROS	
**Quercetin**	Natural flavonoid found in many different plant spp.	MCF-7	50 μg/mL	48 h	↓ Cell viability	↑ after Nec-1 treatment	[130]
↓ Cell proliferation	↑ after Nec-1 treatment
/	/	↑ RIP1, ↑ RIP3 mRNA levels	↓ after Nec-1 treatment
**Resibufogenin**	Asiatic toad (*Bufo gargarizans*)	SW480	5, 10, 15 and 20 μM	24 and 48 h	↑ Necrotic cells		[131]
5, 10 and 20 μM	/	↑ RIP3 protein expression	
10 μM	↑ RIP3 mRNA levels	
HCT116	5, 10 and 20 μM	24 and 48 h	↓ Cell viability	↑ after NAC treatment
↑ after RIP3 knockdown
↑ after NSA treatment
5, 10, 15 and 20 μM	↑ Necrotic cells	↑ after NAC treatment
↑ after RIP3 knockdown
↑ after NSA treatment
20 μM	/	Extensive vacuolation	
Organelle and cell swelling	
10 and 20 μM	24 h	↑ LDH release	
5, 10 and 20 μM	↑ ROS	
5, 10 and 20 μM	/	↑ RIP1 protein expression	
↑ RIP3 protein expression	↓ after RIP3 knockdown
10 μM	↑ RIP3 mRNA levels	
10 and 20 μM	↑ MLKL, ↑ p-MLKL protein expression	↓ after RIP3 knockdown
5 μM	36 h	↓ Cell migration	
↓ Cell invasion	
RIP3 ^+/+^ MEFs	5, 10 and 20 μM	/	↑ PYGL, ↑ GLUL, ↑ GLUD1 protein expression	
↑ PYGL, ↑ GLUL, ↑ GLUD1 activity	
10 μM	↓ Cell migration	
5, 10 and 20 μM	↑ ZO-1, ↑ E-cadherin, ↑ fibronectin, ↑ vimentin, ↑ SNAIL protein expression	
**Sanguilutine**	Plant spp. of *Papaveraceae*, *Fumariaceae*, *Ranunculaceae* and *Rutaceae* families	Mel-JuSo	0.7 μg/mL	48 h	↓ Cell viability	↑ after Nec-1 treatment	[132]
↓ after 3-MA treatment
↓ after LY294002 treatment
A375	0.5, 0.7 and 1 μg/mL	↓ Cell viability	↑ after Nec-1 treatment
↓ after 3-MA treatment
↓ after LY294002 treatment
↓ after BafA1 treatment
↑ after Nec-1 + 3-MA treatment
A375-Bcl2	0.7 μg/mL	↓ Cell viability	↑ after Nec-1 treatment
↓ after 3-MA treatment
↓ after LY294002 treatment
**Shikonin**	*Lithospermum erythrorhizon* Siebold & Zucc., *Arnebia euchroma* (Royle) Johnst, or *Arnebia guttata* Bunge	F-47D	5 μM	12 h	↑ Necrotic cells	↓ after Nec-1 treatment	[133]
4 h	↑ ROS	
AsPC-1	5 and 10 μM	24 h	↑ Necrotic cells	↑ after Nec-1 + Z-VAD-FMK treatment	[134]
↓ after RIP3 knockdown
5 μM	↑ RIP3 mRNA levels	
↑ RIP3 protein expression	
/	24, 48 and 72 h	↓ Cell proliferation	↑ after RIP3 knockdown
CNE-2Z	6.4 μM	24 h	↑ Cell death	↓ after Nec-1 treatment	[135]
6.4 and 12.8 μM	↑ RIP1, ↑ RIP3 protein expression	↓ after Nec-1 treatment
6.4 and 12.8 μM	6 h	↑ ROS	↓ after NAC treatment
/	↑ Mitochondrial ROS	↓ after Nec-1 treatment
3.2, 6.4 and 12.8 μM	/	↓ Cell viability	↑ after Nec-1 treatment
↑ after NAC treatment
A549	3 and 6 μM	3 and 6 h	↓ Cell viability	↑ after Nec-1 treatment	[136]
3 h	↑ Necrotic cells	↓ after Nec-1 treatment
↑ after Z-VAD-FMK treatment
↑ after 3-MA treatment
↑ after BafA1 treatment
↑ after ATG5 siRNA treatment
↑ RIP1 protein expression	↓ after Nec-1 treatment
↑ after Z-VAD-FMK treatment
↑ after 3-MA treatment
↑ after BafA1 treatment
↑ after ATG5 siRNA treatment
KMS-12-PE, RPMI-8226, U266	10 and 20 μM	7 h	↑ Cell death	↓ after Nec-1 treatment	[137]
RPMI-8226	10 μM	2 h	Cell membrane swelling	
Translucent cytoplasm	
20 μM	/	No caspase-3/-8 cleavage	
No RIP1 cleavage	
K7, K12, K7M3, U20S, 143B	3 μM	8 h	↓ Cell viability	↑ after Nec-1 treatment	[138]
K7	3 μM	↑ PI positive cells	↓ after Nec-1 treatment
K7	1, 3 and 5 μM	↑ RIP1, ↑ RIP3 protein expression	
U20S	1, 3, 5 and 7.5 μM	
K7U20S	1, 3 and 5 μM1, 3, 5 and 7.5 μM	No PARP cleavage	

No caspase-3/-6 cleavage	

U937	10 μM	3 and 6 h	↑ LDH release	↓ after Nec-1 treatment	[139]
6 h	↑ Necrotic cells	↓ after Nec-1 treatment
No caspase-3/-8 cleavage	↑ after Nec-1 treatment
↑ TNF-α gene expression	
↑ TNF-α mRNA levels	
↑ TNF-α protein expression	
C6U87	3 and 6 μM 5 and 10 μM	1.5 and 3 h	↓ Cell viability	↑ after Nec-1 treatment	[140]
↑ after NAC treatment
3 h	↑ Necrotic cells	↓ after Nec-1 treatment
↓ after NAC treatment
↑ ROS	↓ after Nec-1 treatment
↓ after NAC treatment
↑ RIP1 protein expression	↓ after Nec-1 treatment
↓ after NAC treatment
Electron-lucent cytoplasm	
Loss of membrane integrity	
Intact nuclear membrane	
Swollen organelles	
U87C6	5 and 10 μM 3 and 6 μM	3 h	↑ LDH release	↓ after Nec-1 treatment	[141]
↓ after GSK-872 treatment
↓ after MnTBAP treatment
↑ ROS	↓ after Nec-1 treatment
↓ after GSK-872 treatment
↓ after MnTBAP treatment
2 h	↑ Mitochondrial O_2_^−^	↓ after Nec-1 treatment
↓ after GSK-872 treatment
↓ after MnTBAP treatment
/	↑ RIP1, ↑ RIP3 protein expression	↓ after Nec-1 treatment
SHG-44U251	2 and 4 μM 5 and 10 μM	3 h	↑ LDH release	↓ after Nec-1 treatment
↓ after GSK-872 treatment
↓ after MnTBAP treatment
↑ after rotenone treatment
↓ after rotenone + Nec-1 treatment
↑ Necrotic cells	↓ after Nec-1 treatment
↓ after GSK-872 treatment
↓ after MnTBAP treatment
↑ after rotenone treatment
↓ after rotenone + Nec-1 treatment
↑ ROS	↓ after Nec-1 after treatment
↓ after GSK-872 treatment
↓ after MnTBAP treatment
↑ after rotenone treatment
↓ after rotenone + Nec-1 treatment
2 h	↑ Mitochondrial O_2_^−^	↓ after Nec-1 treatment
↓ after GSK-872 treatment
↓ after MnTBAP treatment
↑ after rotenone treatment
↓ after rotenone + Nec-1 treatment
/	↑ RIP1 protein expression	↓ after Nec-1 treatment
↑ after rotenone treatment
↓ after MnTBAP treatment
↑ RIP3 protein expression	↓ after GSK-782 treatment
↑ after rotenone treatment
↓ after MnTBAP treatment
RIP1-RIP3 interaction	↑ after rotenone treatment
↓ after MnTBAP treatment
U87C6	5 and 10 μM 3 and 6 μM	3 h	↓ Cell viability	↑ after Nec-1 treatment	[142]
↑ after NAC treatment
15, 30, 60 and 120 min	↑ RIP1 protein expression	↓ after Nec-1 treatment
↓ after RIP1 knockdown
↑ RIP3 protein expression	↓ after Nec-1 treatment
↓ after RIP3 knockdown
U87C6	10 μM 6 μM	15, 30, 60 and 120 min	↑ γ-H2AX protein expression	
↑ p-ATM protein expression	
↑ ROS	↓ after Nec-1 treatment
↓ after GSK-872 treatment
↓ after NAC treatment
1 h	↑ Mitochondrial O_2_^−^	↓ after Nec-1 treatment
↓ after GSK-872 treatment
↓ GSH	↑ after Nec-1 treatment
↑ after GSK-872 treatment
SHG-44U251	4 μM 10 μM	3 h	↑ Necrotic cells	

SHG-44U251	2 and 4 μM 5 and 10 μM	↓ Cell viability	↑ after Nec-1 treatment
↑ after GSK-782 treatment
↑ after RIP1 knockdown
↑ after RIP3 knockdown
↑ after NAC treatment
15, 30, 60 and 120 min	↑ RIP1 protein expression	↓ after Nec-1 treatment
↓ after RIP1 knockdown
↓ after NAC treatment
↑ RIP3 protein expression	↓ after GSK-782 treatment
↓ after RIP3 knockdown
↓ after NAC treatment
SHG-44	4 μM	2 h	RIP1-RIP3 interaction	↓ after Nec-1 treatment
↓ after NAC treatment
SHG-44U251	4 μM 10 μM	15, 30, 60 and 120 min	↑ CypA protein expression	

2 h	No caspase-3/-8 cleavage	
↓ TNF-α release	
↑ TNF-α gene expression	
1 h	↑ DNA damage, ↑ DNA DSBs	↓ after Nec-1 treatment
↓ after GSK-782 treatment
↓ after NAC treatment
↑ γ-H2AX foci	
SHG-44U251	2 and 4 μM 5 and 10 μM	15, 30, 60 and 120 min	↑ γ-H2AX, ↑ p-ATM protein expression	↑ after Nec-1 treatment
↑after GSK-782 treatment
↑ after RIP1 knockdown
↑ after RIP3 knockdown
↓ after NAC treatment
SHG-44U251	4 μM 10 μM	15, 30, 60 and 120 min	↑ ROS	↓ after Nec-1 treatment
↓ after GSK-872 treatment
↓ after NAC treatment
1 h	↑ Mitochondrial O_2_^−^	↓ after Nec-1 treatment
↓ after GSK-872 treatment
↓ GSH	↑ after Nec-1 treatment
↑ after GSK-872 treatment
SHG-44U251	2 and 4 μM 5 and 10 μM	3 h	↑ LDH release	↓ after NSA treatment	[143]
↓ after AIF knockdown
/	↑ Necrotic cells	↓ after NSA treatment
↓ after MLKL knockdown
SHG-44U251U87C6	2 and 4 μM 5 and 10 μM 10 μM 6 μM	15, 30, 60 and 120 min	↑ MLKL, ↑ p-MLKL protein expression	↓ after NSA treatment
↓ after MLKL knockdown


2 h	↓ Mitochondrial AIF protein expression	



SHG-44U251U87C6	4 μM 10 μM 10 μM 6 μM	2 h	↑ Cytoplasmatic/nuclear AIF protein expression	↓ after AIF knockdown
↓ after NSA treatment
↓ after MLKL knockdown
↓ after MnTBAP treatment


SHG-44U251	4 μM 10 μM	2 h	↓ Δψm	↑ after NSA treatment
↑ after MnTBAP treatment
/	↑ Mitochondrial O_2_^−^	↓ after NSA treatment
↓ after MnTBAP treatment
↓ after MLKL knockdown
/	↑ ROS	↓ after MnTBAP treatment
↓ after MLKL knockdown
15, 30, 60 and 120 min	↑ Mitochondrial MLKL, ↑ Mitochondrial p-MLKL protein expression	
SHG-44	4 μM	2 h	Mitochondrial accumulation of MLKL	
5–8F	7.5 μM	6 h	Ruptured plasma membrane		[144]
Increased cell volume	
Swollen organelles	
Loss of membrane integrity	
↑ Necrotic cells	↓ after Nec-1 treatment
↓ after NAC treatment
↑ Caspase-3/-8 activity	↑ after Nec-1 treatment
/	↑ ROS	↓ after NAC treatment
/	↑ RIP1, ↑ RIP3, ↑ MLKL protein expression	↓ after Nec-1 treatment
↓ after NAC treatment
AGS	1, 2 and 4 μM	6 h	↑ Necrotic cells	↓ after Nec-1 treatment	[145]
↑ ROS	↓ after NAC treatment
↓ Δψm	↑ after Nec-1 treatment
↑ after NAC treatment
2 and 4 μM	↓ Cell viability	↑ after Nec-1 treatment
NIH3T3	0.5, 1 and 2.5 μM	3 h	↓ Cytotoxicity	↓ after Nec-1 treatment	[146]
↑ ROS	↓ after NAC treatment
/	RIP1-RIP3 interaction	↓ after Nec-1 treatment
MCF-7	5 μM	12 h	↑ Necrotic cells	↓ after Nec-1 treatment	[147]
↑ RIP1, ↑ RIP3 protein expression	
4 h	↑ ROS	
12 h	↓ Δψm	↓ after Nec-1 treatment
24 h	No caspase-3 activation	
↑ Caspase-8 activity	↑ after Nec-1 treatment
12 h	Extensive vacuolation	
Loss of membrane integrity	
MDA-MB-468	5 μM	12 h	↑ Necrotic cells	↓ after Nec-1 treatment	[148]
↑ RIP1, ↑ RIP3 protein expression	
4 h	↑ ROS	
12 h	↓ Δψm	↓ after Nec-1 treatment
24 h	↑ Caspase-3/-8 activity	↑ after Nec-1 treatment
12 h	Extensive vacuolation	
Loss of membrane integrity	
**Tanshinol A**	*Salvia miltiorrhiza* Bunge	H1299	2, 4 and 8 μM	8 h	No caspase-3/-7 activation		[149]
1, 2, 4 and 8 h	↑ p-MLKL protein expression	
8 μM	8 h	↑ PI positive cells	
↓ Cell viability	↑ after MLKL knockdown
↑ after NAC treatment
↑ after CAT treatment
↓ after MLKL transfection
↑ MLKL membrane translocation	
↑ ROS	↓ after NAC treatment
↓ after CAT treatment
A549	10, 15 and 20 μM	8 h	No caspase-3/-7 activation	
No caspase-3/-7/-8/-9 cleavage	
20 μM	/	↑ PI positive cells	
15 and 20 μM	↑ Necrotic cells	
10, 15 and 20 μM	1, 2, 4 and 8 h	↑ p-MLKL protein expression	↓ after MLKL knockdown
↓ after NAC treatment
↓ after CAT treatment
20 μM	8 h	↑ MLKL membrane translocation	↓ after NAC treatment
↓ Cell viability	↑ after MLKL knockdown
↑ after NAC treatment
↑ after CAT treatment
↓ after MLKL transfection
↑ ROS	↓ after NAC treatment
↓ after CAT treatment
**Tanshinone IIA**	*Salvia miltiorrhiza* Bunge	HepG2	5 and 10 μg/mL	12 h	↑ Necrotic cells	↓ after Nec-1 treatment	[150]
↑ after Z-VAD-FMK treatment
↑ LDH release	↓ after Nec-1 treatment
↑ after Z-VAD-FMK treatment
↑ CypA protein expression	↓ after Nec-1 treatment
↑ after Z-VAD-FMK treatment
↑ HMGB1 protein expression	↓ after Nec-1 treatment
↑ after Z-VAD-FMK treatment
↑ FLIP_L_, ↑ FLIP_S_ protein expression	↑ after Nec-1 treatment
↑ Cleaved caspase-3/-8 protein expression	↓ after Nec-1 treatment
↓ after Z-VAD-FMK treatment
↓ RIP1 protein expression	↑ after Z-VAD-FMK treatment
↑ Cleaved RIP1 protein expression	↓ after Nec-1 treatment
↓after Z-VAD-FMK treatment
Formation of RIP1/RIP3/FADD/FLIP_s_ complex	
↓ MLKL monomer protein expression	↑ after Nec-1 treatment
↑ after Z-VAD-FMK treatment
↑ Cleaved MLKL protein expression	↓ after Nec-1 treatment
↓ after Z-VAD-FMK treatment
**Ungeremine**	*Crinum zeylanicum* L.	CCRF-CEM	2.37, 3.76, 9.40, 18.79, 37.58, 75.17 and 150.33 μM	24 h	↓ Cell proliferation	↑ after Nec-1 treatment	[53]
1.22, 2.45, 4.89 and 9.78 μM	↑ ROS	
4.89 and 9.78 μM	↑ RIP3 protein expression	
**Youdujing extract**	Traditional Chinese herbal formula Youdujing	Ect1/E6E7	1, 2 and 4 mg/mL	12 h	No caspase-3/-7 activation		[151]
2 and 4 mg/mL	/	↑ RIP1 protein expression	↓ after RIP1 knockdown
↑ after RIP1 overexpression
4 mg/mL	RIP1-MLKL interaction	
3 h	↓ Cell viability	↑ after Nec-1 treatment
↑ after RIP1 knockdown
↓ after RIP1 overexpression

*Abbreviations*: ↑: Increase; ↓: Decrease; 3-MA: 3-methyladenine; A549/Cis: cisplatin resistant A549 cells; A375-Bcl2: Bcl-2 transfected A-375 cells; AIF: Apoptosis-inducing factor; Akt: Protein kinase B; AR: Androgen receptor; ATG5: Autophagy-related gene 5; ATP: Adenosine triphosphate; BafA1: Bafilomycin-A1; BAPTA-AM: Calcium chelator; BGN: Biglycan; Ca^2+^: Calcium; CaMKII: Calcium-calmodulin dependent protein kinase II dihydroethidium; CAT: Catalase; CCN1: Cell communication network factor 1; CQ: Chloroquine; CypA: Cyclophilin A; DI60N: Kinase dead RIP3; DIC: Dicoumarol; DSBs: Double strand-breaks; DTT: Dithiothreitol; EGF: Epidermal growth factor; FADD: FAS-associated death domain; FasL: FAS ligand; FLIP_L_: Cellular FLICE (FADD-like IL-1β-converting enzyme)-inhibitory protein isoform L; FLIP_S_: Cellular FLICE (FADD-like IL-1β-converting enzyme)-inhibitory protein isoform S; GLUD1: Glutamate dehydrogenase; GLUL: Glutamine synthetase; GLUT1/4: Glucose transporter 1/4; GSH: Glutathione; GSK-872: RIP3 inhibitor; GSSG: Glutathione disulfide; H_2_O_2_: Hydrogen peroxide; Hgb: Hemoglobin; HK2: Hexokinase 2; HROS: Highly Reactive oxygen species; IL-6: Interleukin-6; IL-8: Interleukin-8; iNOS: Inducible nitric oxide synthase; JNK1/2: c-Jun N-terminal kinase 1/2; K45A: Kinase dead RIP1; KN93: CaMKII inhibitor; L-NMMA: NG-monomethyl L- arginine, pan-NOS inhibitor; LDH: Lactate dehydrogenase; LY294002: Autophagy inhibitor; MLKL: Mixed lineage kinase domain like pseudokinase; MnSOD: Manganese superoxide dismutase; MnTBAP: Superoxide dismutase mimetic and peroxynitrite scavenger; NAC: N-acetyl-L-cysteine; Nec-1: Necrostatin-1; Nec-1s: Necrostatin-1s, 7-Cl-O-Nec-1; NF-kB: Nuclear factor kappa-light-chain-enhancer of activated B cells; NO: Nitric oxide; NQO1: NAD(P)H: quinone oxidoreductase 1; NSA: Necrosulfonamide; O_2_^−^: Superoxide; p-Akt: Phospho-protein kinase B; p-ATM: Phospho-ataxia telangiectasia mutated kinase; p-ATR: Phospho-ataxia telangiectasia and Rad3-related kinase; p-CHK1/2: Phospho-checkpoint kinase 1/2; p-IκBα: Phospho-inhibitor of nuclear factor kappa B; p-MKLK: Phospho-mixed lineage kinase domain like pseudokinase; p-mTOR: Phospho-mammalian target of rapamycin; p-p70S6K1: Phospho-p70 ribosomal protein S6 kinase 1; p-RIP1: Phospho-receptor-interacting serine/threonine-protein kinase 1; p-RIP3: Phospho-receptor-interacting serine/threonine-protein kinase 3; p-TSC2: Phospho-tuberous sclerosis complex 2; PARP: Poly ADP (adenosine diphosphate)-ribose polymerase; PCYT1A: Phosphate cytidylyltransferase 1 alpha; PI: Propidium iodide; POCCLs: Patient-derived primary ovarian cancer cell lines; PSA: Prostate specific antigen; PYGL: Glycogen phosphorylase; RIP1: Receptor-interacting serine/threonine-protein kinase 1; RIP3: Receptor-interacting serine/threonine-protein kinase 3; ROS: Reactive oxygen species; SB203580: P38 inhibitor; SOD-1/2: Superoxide dismutase 1/2; Spp.: species; SP600125: JNK inhibitor; TNF-α: Tumor necrosis factor-α; z-LLY-fmk: Calpain inhibitor; Z-VAD-FMK: pan-caspase inhibitor; ZO-1: Zonula occludens-1; γ-H2AX: Phospho-H2A histone family member X; ΔΨm: Mitochondrial membrane potential.

**Table 3 cancers-13-00304-t003:** Natural products as in vitro inducers of pyroptosis.

Compound	Compound Source	Cell Line (s)	Concentrations (Where Specified)	Time (Where Specified)	Pyroptosis Markers	Supplementary Effects	Reference
Alpinumisoflavone	*Derris eriocarpa* F.C	KYSE30, KYSE510	5, 10 and 20 μM	24 and 48 h	↓ Cell viability	↓ after GSDME knockdown	[201]
			10 and 20 μM	/	↓ Colony formation		
					Cell swelling and bubble at plasma membrane		
					↑ LDH release	↓ after Z-DEVD-FMK treatment	
						↓ after caspase-3 knockdown	
						↓ after GSDME knockdown	
					↑ Cleaved GSDME protein expression	↓ after Z-DEVD-FMK treatment	
						↓ after caspase-3 knockdown	
		Huh7, MMC 7721	2, 5, 10 and 20 μM	48 h	↓ Cell viability	↓ after MCC950 treatment	[202]
						↓ after NLPR3 knockdown	
			10 and 20 μM	14 days	↓ Colony formation		
				24 h	↓ Cell invasion	↑ after MCC950 treatment	
						↑ after NLPR3 knockdown	
					↓ Cell migration	↑ after MCC950 treatment	
						↑ after NLPR3 knockdown	
				48 h	↑ LDH release	↑ after CQ treatment	
						↑ after ATG5 knockdown	
					↑ NLRP3, ↑ cleaved caspase-1, ↑ cleaved IL-1β, ↑ cleaved IL-18 mRNA levels		
					↑ NLRP3, ↑ cleaved caspase-1, ↑ cleaved IL-1β, ↑ cleaved IL-18, ↑ cleaved GSDMD protein expression	↓ after MCC950 treatment	
						↓ after NLPR3 knockdown	
						↑ after CQ treatment	
						↑ after ATG5 knockdown	
Anthocyanin	Flavonoid found in different plant spp.	Tca8113, SCC15	250 μg/mL	24, 48 and 72 h	↓ Cell viability	↑ after AC-YVAD-CMK treatment	[203]
				48 h	↓ Cell migration	↑ after AC-YVAD-CMK treatment	
				48 h	↓ Cell invasion	↑ after AC-YVAD-CMK treatment	
				48 h	↑ NLRP3, ↑ caspase-1, ↑ IL-1β mRNA levels	↓ after AC-YVAD-CMK treatment	
				/	↑ NLRP3, ↑ cleaved caspase-1, ↑ cleaved IL-1β, ↑ cleaved IL-18 protein expression	↓ after AC-YVAD-CMK treatment	
				/	↑ GSDMD protein expression		
Berberine	Huang Lian Chinese herb *(Coptis chinesis)* and others plant spp.	HepG2	50 and 100 μM	/	Cell swelling		[204]
			50 μM	24, 48 and 72 h	↓ Cell viability	↑ after AC-YVAD-CMK treatment	
				48 h	↓ Cell migration	↑ after AC-YVAD-CMK treatment	
				48 h	↓ Cell invasion	↑ after AC-YVAD-CMK treatment	
			25, 50 and 100 μM	/	↑ Caspase-1 mRNA levels	↓ after AC-YVAD-CMK treatment	
					↑ Caspase-1 protein expression	↓ after AC-YVAD-CMK treatment	
Casticin	*Vitex* spp.	5–8F	/	24 h	↑ LDH release	↓ after SP60012 treatment	[205]
						↓ after JSH-23 treatment	
						↓ after PKR knockdown	
			3, 6 and 9 μM		↑ Cleaved caspase-1, ↑ cleaved GSDMD, ↑ PKR, ↑ IL-1β, ↑ p-NF-κB, ↑ p-JNK protein expression	↓ after SP600125 treatment	
						↓ after JSH-23 treatment	
						↓ after PKR knockdown	
			6 μM		↑ IL-6, ↑ IL-1β, ↑ TLR4, ↑ ASC mRNA levels		
Dioscin	*Polygonatum zanlanscianense* Pamp., *Dioscorea nipponica* Makino, and *Dioscorea zingiberensis* C. H. Wright	MNNG/HOS	/	24 h	Bubbles at plasma membrane		[206]
			2.5 and 5 μM		↑ LDH release		
					↑ Cleaved GSDME protein expression		
		MG63	/		Bubbles at plasma membrane		
			2 and 4 μM		↑ LDH release		
					↑ Cleaved GSDME protein expression		
		U20S	/		Bubbles at plasma membrane		
			2 and 4 μM		↑ LDH release	↓ after Z-DEVD-FMK treatment	
					↑ Cleaved GSDME protein expression	↓ after Z-DEVD-FMK treatment	
						↓ after GSDME knockdown	
			4 μM		↓ Cell viability	↑ after Z-DEVD-FMK treatment	
						↑ after SPSP600125 treatment	
Galangin	*Alpinia officinarum* Hance, *Alnus pendula* Matsum, *Plantago major* L, and *Scutellaria galericulata* L.	U251, U87MG	150 μM	48 h	Bubbles at plasma membrane		[207]
			/	12, 24, 48 and 72 h	↑ LDH release	↓ after GSDME knockdown	
			150 μM	48 h	↑ Cleaved GSDME protein expression	↑ after 3-MA treatment	
Huaier extract	*Trametes robiniophila* Murr (Huaier)	H520, H358	5 and 10 mg/mL	24 and 48 h	↓ Cell viability	↑ after NLRP3 knockdown	[208]
						↑ after MCC950 treatment	
					↑ LDH release	↓ after NLRP3 knockdown	
						↓ after MCC950 treatment	
					↑ NLRP3, ↑ caspase-1, ↑ IL-1β, ↑ IL-18 mRNA levels		
					↑ NLRP3, ↑ cleaved caspase-1, ↑ cleaved IL-1β, ↑ cleaved IL-18 protein expression	↓ after NLRP3 knockdown	
						↓ after MCC950 treatment	
L50377 (piperlongumine analogue)	*Piper Longum* L.	A549	10 μM	/	Cell swelling and bubbles at plasma membrane	↓ after NAC treatment	[209]
						↓ after IKKβ overexpression	
			0.74, 2.22, 6.67 and 20 μM		↓ Cell viability	↓ after NAC treatment	
						↓ after IKKβ overexpression	
			5, 10 and 20 μM	2 h	↓ IKKα, ↓ IKKβ phosphorylation		
			2.5 μM	8 h	↑ ROS	↓ after NAC treatment	
			5 and 10 μM	/	↑ Cleaved GSDME protein expression		
Nobiletin	*Citrus* fruits	A2780, OVCAR-3	10, 20, 30, 40 and 50 μM	24 h	↓ Cell viability		[210]
			50 μM	12 h	↑ ROS	↓ after NAC treatment	
			10, 30 and 50 μM		↓ ΔΨm		
			50 μM	24 h	Cell swelling and bubbles at plasma membrane		
			10, 30 and 50 μM		↑ Cleaved GSDMD protein expression	↓ after NAC treatment	
						↓ after 3-MA treatment	
					↑ Cleaved GSDME protein expression	↓ after NAC treatment	
						↓ after 3-MA treatment	
						↓ after NSA treatment	
			50 μM		↑ IL-1β, ↑ ASC mRNA levels		
Osthole	*Cnidium monnieri* L. Cusson	A2780, OVCAR-3	10, 20, 30, 40, 50, 60, 70 and 80 μM	24 h	↓ Cell viability		[211]
			20, 40 and 80 μM		Cell swelling and bubbles at plasma membrane		
					↑ Cleaved GSDME protein expression		
Paclitaxel	Pacific yew	A549	20 and 60 μM	24 and 48 h	↑ Lytic cell death	↓ after AC-DEVD-CHO treatment	[212]
						↓ after GSDME knockdown	
			60, 120, 180 and 240 μM		↑ Cleaved caspase-3, ↑ cleaved caspase-7, ↑ cleaved caspase-8, ↑ cleaved caspase-9, ↑ cleaved GSDME protein expression		
Polyphyllin VI	*Trillium tschonoskii* Maxim	A549, H1299	3, 4, 5 and 6 μM	24 h	↑ Pyroptotic cells	↓ after VX-765 treatment	[198]
						↓ after NSA treatment	
			4 μM		↑ Activated caspase-1 expression		
					↑ PI positive cells	↓ after VX-765 treatment	
						↓ after BAY treatment	
			3, 4, 5 and 6 μM		↑ NLRP3, ↑ cleaved caspase-1, ↑ cleaved IL-1β, ↑ cleaved IL-18, ↑ cleaved GSDMD protein expression		
					↑ IL-1β, ↑ IL-18 secretion		
					↑ ROS	↓ after NAC treatment	
					↑ p65/NF-kB protein expression	↓ after NAC treatment	
Tanshinone IIA	*Salvia miltiorrhiza Bunge* (Danshen)	HeLa	2, 4 and 8 μM	24, 48 and 72 h	↓ Cell proliferation		[213]
				72 h	↑ IL-1β, ↑ IL-18, ↑ GSDMD protein expression	↓ after miR-145 knockdown	

*Abbreviations*: ↑: Increase; ↓: Decrease; 3-MA: 3-methyladenine; AC-DEVD-CHO: Caspase-3 inhibitor; AC-YVAD-CMK: Caspase-1 inhibitor; ASC: Apoptosis-associated speck-like protein containing a CARD (caspase activation and recruitment domain); ATG5: Autophagy related 5; BAY: Bay 11-7082, NF-kB inhibitor; GSDMD: Gasdermin D; GSDME: Gasdermin E; IKKα: Inhibitor of nuclear factor kappa-B kinase subunit alfa; IKKβ: Inhibitor of NF-kB kinase subunit beta; IL-6: Interleukin-6; IL-18: Interleukin-18; IL-1β: Interleukin 1 beta; JSH-23: NF-κB inhibitor; LDH: Lactate dehydrogenase; MCC950: NLRP3 inhibitor; NAC: N-acetylcysteine; NF-kB: Nuclear factor kappa-light-chain-enhancer of activated B cells; NLRP3: NLR (nucleotide-binding oligomerization domain (NOD)-like receptor) family pyrin domain-containing 3; NSA: Necrosulfonamide; PKR: Protein kinase R; p-JNK1: Phospho-c-Jun N-terminal kinase; p65: Transcription factor p65; Spp.: species; SPSP600125: Inhibitor of c-Jun N-terminal kinase (JNK); VX-765: Caspase-1 inhibitor; Z-DEVD-FMK: Caspase-3 inhibitor.

## Data Availability

Not applicable.

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
