# Peer review of "Natural Products as Inducers of Non-Canonical Cell Death: A Weapon against Cancer"

_cancers, 2021, doi:10.3390/cancers13020304_

Round 1

Reviewer 1 Report

As non-apoptotic programmed cell death (also called as non-canonical cell death) has been shown to be one of mechanisms of antitumor agents-mediated cancer cell death, in this review Greco et al. summarized non-apoptotic programmed cell death including ferroptosis, necroptosis, and pyroptosis. Additionally, the authors gathered bunch of references of natural products which has been shown to induce non-apoptotic cell death in vitro and in vivo. The authors well searched about natural products. They also discussed well about how apoptotic and non-apoptotic programmed cell death work during cancer treatment. I think this is an impressive and educational review. I have several comments for this review.

  • It would be better to describe non-apoptotic cell death in more detail; what is the definition of each non-apoptotic cell death, what is the differences of the non-apoptotic cell death and apoptotic cell death, and how to detect/differentiate each non-apoptotic cell death.
  • It would be better to add/describe the aim of this review compactly in Abstract.
  • In this review, the authors seem skipped summarizing autophagy in antitumor agent-mediated cancer cell death. Is there chance to add about autophagy? Or is this beyond their scope?
  • Heading number “4.” is overlapped. Conclusion should be “5.”.

Author Response

The authors of this manuscript express their sincere thanks to the reviewers for the critical assessment of this work. The authors have acted upon the recommendations of the reviewers, which have resulted in a significant enhancement in the quality of this manuscript. All modifications incorporated in the manuscript are highlighted in red color font. A “point-by-point” response to each and every comment is outlined below.

Reviewer 1

As non-apoptotic programmed cell death (also called as non-canonical cell death) has been shown to be one of mechanisms of antitumor agents-mediated cancer cell death, in this review Greco et al. summarized non-apoptotic programmed cell death including ferroptosis, necroptosis, and pyroptosis. Additionally, the authors gathered bunch of references of natural products which has been shown to induce non-apoptotic cell death in vitro and in vivo. The authors well searched about natural products. They also discussed well about how apoptotic and non-apoptotic programmed cell death work during cancer treatment. I think this is an impressive and educational review. I have several comments for this review.

  • It would be better to describe non-apoptotic cell death in more detail; what is the definition of each non-apoptotic cell death, what is the differences of the non-apoptotic cell death and apoptotic cell death, and how to detect/differentiate each non-apoptotic cell death.

We thank Reviewer 1 for his/her comment. We expanded the description of the mechanisms of the various non-canonical cell death pathways, including some of the distinctive features of each of them (lines 68-73; 84-98; 258-271; 273-275; 403-407). However, the description of the detection/differentiation methods has not been added as we believe that this methodological information is beyond the scope of this review.

  • It would be better to add/describe the aim of this review compactly in Abstract.

We thank Reviewer 1 for his/her advice and we modified the text as suggested (lines 15-16)

  • In this review, the authors seem skipped summarizing autophagy in antitumor agent-mediated cancer cell death. Is there chance to add about autophagy? Or is this beyond their scope?

We thank Reviewer 1 for pointing out this point. Autophagy, surely, is an important aspect in the evaluation of anticancer activity. However, given the complexity of the autophagic process, we thought that adding further information about the induction of autophagy by naturally occurring compounds is beyond the scope of this review.

  • Heading number “4.” is overlapped. Conclusion should be “5.”

We apologize for the mistake and we corrected the text as suggested (line 612).

Reviewer 2 Report

The font size in Figure 1 should be standardized. Some words are too small to be read. This also applies to other figures. The readability of Table 1 should also be improved. It is hard to be read and is confusing. Some paragraphs are too short to be one paragraph. Authors are suggested to edit the manuscript thoroughly, and improve the organization. The numbering of the section is totally wrong. Why two sections are numbered as “4”? Finally, besides ferroptosis, necroptosis, and pyroptosis, any other mechanisms are involved? Authors are suggested to discuss further, and cover different mechanisms, if any, to enhance the comprehensiveness of the review.

Author Response

The authors of this manuscript express their sincere thanks to the reviewers for the critical assessment of this work. The authors have acted upon the recommendations of the reviewers, which have resulted in a significant enhancement in the quality of this manuscript. All modifications incorporated in the manuscript are highlighted in red color font. A “point-by-point” response to each and every comment is outlined below.

Reviewer 2

  • The font size in Figure 1 should be standardized. Some words are too small to be read. This also applies to other figures. The readability of Table 1 should also be improved.

We thank Reviewer 2 for pointing out these critical points. We changed the font size in figures and tables according to her/his suggestions.

  • It is hard to be read and is confusing. Some paragraphs are too short to be one paragraph. Authors are suggested to edit the manuscript thoroughly, and improve the organization.

We thank Reviewer 2 for pointing out this critical point. We corrected the text as suggested assembling some paragraph (lines 142, 148, 320, 350, 396, 477).

  • The numbering of the section is totally wrong. Why two sections are numbered as “4”?

We apologize for the numbering mistake, which we corrected (line 612).

  • Finally, besides ferroptosis, necroptosis, and pyroptosis, any other mechanisms are involved? Authors are suggested to discuss further, and cover different mechanisms, if any, to enhance the comprehensiveness of the review.

Yes, we agree with Reviewer 2 that other non-apoptotic cell deaths, such as autophagy, anoikis, paraptosis, partanathos, netosis, or entosis could represent interesting tools for fighting cancer. However, since for some of them the mechanism of action is not completely understood or there is a paucity of data about the ability of natural compounds to trigger them, we decided to focus only on the most studied and characterized ferroptosis, necroptosis, and pyroptosis. We added a sentence to clarify our choice (lines 53-59).

Reviewer 3 Report

In the manuscript entitled “Natural products as inducer of non-canonical cell death: a weapon against cancer”, Greco et al.reviewed the effects of Natural products to induce non-canonical cell death. This review paper is fairly well written. I personally enjoyed the manuscript. I have a few minor questions and comments.

Although the effects of many natural products were described, I concern a little about whether those products really induce preferential cancer cell death, or not, compared to normal cells, and how much such kind of comparative studies are carefully done in the study field.

Since killing pathways of cancer cells are usually complex and are tricky, the pathways to induce cell death are often context-dependent, as the authors also mentioned in the manuscript. I am appreciated if the authors could involve that information as much as possible.

Author Response

The authors of this manuscript express their sincere thanks to the reviewers for the critical assessment of this work. The authors have acted upon the recommendations of the reviewers, which have resulted in a significant enhancement in the quality of this manuscript. All modifications incorporated in the manuscript are highlighted in red color font. A “point-by-point” response to each and every comment is outlined below.

Reviewer 3

In the manuscript entitled “Natural products as inducer of non-canonical cell death: a weapon against cancer”, Greco et al. reviewed the effects of Natural products to induce non-canonical cell death. This review paper is fairly well written. I personally enjoyed the manuscript. I have a few minor questions and comments.

  • Although the effects of many natural products were described, I concern a little about whether those products really induce preferential cancer cell death, or not, compared to normal cells, and how much such kind of comparative studies are carefully done in the study field.

We thank Reviewer 3 for his/her appreciated comment. We added a new paragraph where we present the selective induction of cell death in cancer cells (lines 509-609). The information about selectivity previously reported in the text were moved to this new paragraph.

  • Since killing pathways of cancer cells are usually complex and are tricky, the pathways to induce cell death are often context-dependent, as the authors also mentioned in the manuscript. I am appreciated if the authors could involve that information as much as possible.

We thank Reviewer 3 for his/her appreciated comment. In our opinion, we already focused our attention on this aspect both in the main text and in the conclusions, where we explicitly explain that the cell status or the presence of mutations are the knobs that are turned to choose one or the other killing pathway will be triggered (lines conclusions). For instance, we showed that a single compound can trigger different cell death modalities and that the cell-dependent expression of the key molecular mediators involved in the different killing pathways (i. e. RIP1, RIP3, GSDMD, and GSDME) dictates the type of induced cell death (lines 652-665). In the conclusions, we also extensively show how all cell death modalities are intertwined (lines conclusions). See also lines 322-330 for shikonin-induced necroptosis versus apoptosis on MCF-7 based on caspases inhibition; lines 331-335 for cell fate necroptosis versus apoptosis depending on caspase-8/cIAP1/cIAP2; lines 454-462 for GSDM expression on cancer cells as discriminating factor in triggering apoptosis, pyroptosis or both.

Moreover, we described the complex interactions among different types of cell death (lines 134-137; 145-149 for the role played by autophagy in ferroptosis and lines 476-483 for pyroptosis). Finally, please see the involvement of ER stress in ferroptotic cell death and how the induction of ER stress can promote or hinder the induction of ferroptosis depending on the cell model (lines 161-167; 170-172); the interrelation between necroptosis and apoptosis (lines 322-329); the interrelation between pyroptosis and apoptosis (lines 452-462). However, if the Reviewer does not agree with us, we will be happy to add more details.

Round 2

Reviewer 2 Report

The authors have made changes to the manuscript to address reviewers' concerns. I recommend it for publication.